# Towards Understanding the Robustness Against Evasion Attack on Categorical Inputs

**Hongyan Bao**[*]
King Abdullah University of Science and Technology
`hongyan.bao@kaust.edu.sa`

**Yufei Han**[*]
INRIA
`yufei.han@inria.fr`

**Yujun Zhou**
King Abdullah University of Science and Technology
`yujun.zhou@kaust.edu.sa`

**Yun Shen**
NetApp [†]
`yun.shen@netapp.com`

**Xiangliang Zhang** [‡]
University of Notre Dame
`xzhang33@nd.edu`

## Abstract

Characterizing and assessing the adversarial risk of a classifier with categorical inputs has been a practically important yet rarely explored research problem. Conventional wisdom attributes the difficulty of solving the problem to its combinatorial nature. Previous research efforts tackling this problem are specific to use cases and heavily depend on domain knowledge. Such limitations prevent their general applicability in real-world applications with categorical data. Our study novelly shows that provably optimal adversarial robustness assessment is computationally feasible for any classifier with a mild smoothness constraint. We theoretically analyze the impact factors of adversarial vulnerability of a classifier with categorical inputs via an information-theoretic adversarial risk analysis. Corroborating these theoretical findings with a substantial experimental study over various real-world categorical datasets, we can empirically assess the impact of the key adversarial risk factors over a targeted learning system with categorical inputs.

## 1 Introduction

Categorical data pervasively exist in real-world safety-critical Machine-Learning-as-a-Service (MLaaS) applications, such as ML-driven intrusion detection and digital healthcare. The vulnerability to attacks by intentionally crafting categorical signatures raises concerns on trust and utility of the ML-based analytic services. Characterizing and assessing adversarial robustness on categorical data can thus help evaluate the reliability of the core ML models and flag potential evading efforts. For a classifier $f$ with categorical inputs $x$, the adversarial risk of $f$ under evasion attack can be formulated as follows.

**Definition 1** $f : x \rightarrow y$ *denotes a classifier with categorical inputs $x$. Let $\mu_{x,y}$ denote the joint distribution of $(x,y)$. The expected adversarial risk of the classifier $f$ under evasion attack is formulated as:*

$$\mathcal{R}_{\varepsilon}^{adv} = \mathop{\mathbb{E}}_{(x,y) \sim \mu_{x,y}} \sup_{|diff(x,\hat{x})| \leq \varepsilon} \ell(f(\hat{x}), y) \tag{1}$$

*where $\ell$ is the misclassification loss function, $x$ and $\hat{x}$ are an unperturbed and the generated adversarial sample respectively. $\varepsilon$ denotes the attack budget of evasion attack. $x$ is correctly classified ($f(x) = y$).*

Intuitively, a classifier $f$ is more robust against adversarial perturbations, if its adversarial risk is low given the attack budget limiting the number of changed categorical features. Unlike continuous data, a categorical variable can be valued with only one categorical value among others. These categorical

---

[*]The first two authors contributed equally.

[†]The author contributed to this work while at NortonLifeLock.

[‡]Corresponding author. Dr. Zhang is also affiliated with King Abdullah University of Science and Technology

values have no intrinsic ordering to the categories. Evasion attack manipulating categorical inputs is in nature **an NP-hard knapsack problem**. The discontinuous nature raises two fundamental yet rarely addressed questions to evaluate the adversarial risk on categorical data in practice:

**Q1** What are the key factors determining $f$'s adversarial risk $\mathcal{R}_\varepsilon^{adv}$ on categorical data ?

**Q2** For a general classifier $f$, can we assess the adversarial risk of $f$ with categorical inputs with provably accuracy guarantee ?

Despite recent efforts of adversarial vulnerability exploration with discrete data, both questions remain open for several reasons. First, the discontinuity of categorical space prevents the direct use of the previous progress on adversarial risk analysis with continuous data. The local subspace assumption of $l_p$-bounded adversarial attacks does not apply to the categorical features (Hein & Andriushchenko, 2017; Wang et al., 2018; Fawzi et al., 2016; Gilmer et al., 2018; D.Yin et al., 2019; Khim & Loh, 2018; Tu et al., 2019). Second, most practices of discrete adversarial attack are domain specific and depend heavily on domain knowledge. (Bojchevski & Günnemann, 2019; Bojchevski & Günnemann, 2019; Zugner & Gunnemann, 2019) focus on building differentiable surrogate functions to Graph Neural Networks to facilitate searching for feasible poisoning edits over graph structures and node attributes. (Narodytska & Kasiviswanathan, 2017; Croce & Hein, 2019) conduct $L_0$-norm perturbations only within local image areas containing sensitive features for image classification. (Qi et al., 2019; Wang et al., 2020) require non-negativity on the parameters of deep neural networks to deliver provably accurate greedy attacks via submodular function optimization. The non-negativity constraint is unnatural for real-world ML practices. It does harm the utility of the classifier. For a more general classifier with categorical inputs, a provably optimal and domain-agnostic method for attack and adversarial risk evaluation is yet to establish. Using greedy search or the well-known Branch-and-Bound method on a general knapsack problem provides no optimality guarantee of the solutions, thus can produce arbitrarily bad results.

Our study aims to address these two questions mentioned above from both theoretical and empirical perspectives. **First**, we derive an information-theoretic characterization of the adversarial risk of a classifier. It unveils that *the informativeness of the input categorical instance*, *the sensitivity of the perturbed categorical features* and *the information geometry property of the targeted classifier* are the three key factors jointly determining adversarial vulnerability of the classifier.

**Second**, our study adopts an assess-by-attack strategy. We show that assessing adversarial robustness of *any measurable classifier with categorical inputs* can be cast to a weakly submodular maximization problem, with a mild smoothness condition. It can thus be solved using a simple yet efficient greedy attack strategy with provable approximation guarantees. *The theoretical findings not only explain the empirical success of the greedy search strategy to generate adversarial textual and image samples (Gong et al., 2018; Yang et al., 2018; Narodytska & Kasiviswanathan, 2017), but also pave the way to a domain-agnostic adversarial robustness assessment on categorical data with provable optimality guarantees.*

**Third**, we instantiate the domain-agnostic adversarial risk characterization and assessment with a widely used DNN classifier, i.e. *Long Short-Term Memory* (LSTM) and three different categorical datasets in Section.4. These datasets are collected from various real-world applications. The experimental results confirm the impact of the 3 risk factors over the adversarial vulnerability of the DNN classifier.

## 2 RELATED WORK

Tremendous efforts have been made to *vulnerability measurement of a classifier under evasion attack* (Hein & Andriushchenko, 2017; Wang et al., 2018; Fawzi et al., 2016; Gilmer et al., 2018; Weng et al., 2019; Sinha et al., 2018; Cohen et al., 2019; Shi et al., 2020; D.Yin et al., 2019; Khim & Loh, 2018; Tu et al., 2019). Most of the previous works focus on evaluating robustness against $l_p$-norm perturbations on continuous data. They all assume adversarial samples locate within a smooth $l_p$-ball around an input instance, which doesn't hold for categorical data. In contrast, Tu et al. (2019) covers both numerical and categorical data. It bounds the adversarial risk with a local worst-case risk over a $p$-wassernstein ball centered at the training data distribution. This work associates the adversarial risk of a classifier with its rademacher complexity. Nevertheless, we argue that the adversarial vulnerability of a classifier is determined by not only the characteristics of the classifier, e.g. model complexity, but also the properties of the training/testing data instances.

Pioneering works of evasion attacks with categorical inputs depend on domain-specific knowledge to facilitate the attack exploration. Kuleshov et al. (2018); Papernot et al. (2016); Miyato et al. (2016); Samanta & Mehta (2017); Gao et al. (2018); Yang et al. (2018); Gong et al. (2018); Ebrahimi et al.

(2018); Narodytska & Kasiviswanathan (2017); Croce & Hein (2019) focus on replacing individual words/phrases to cheat text classifiers, or modifying pixel intensities to bias image classification results. These methods use heuristic semantic rules, e.g., replacing words with manually defined candidate synonyms and constraining the word change to preserve readability and semantic integrity. Narodytska & Kasiviswanathan (2017); Croce & Hein (2019) narrow down the search range to the pixels with high pixel-wise sensitivity for image classification. Despite of the sounding empirical results, there is no guarantee on a successful attack within the attack budget.

Bojchevski & Günnemann (2019); Bojchevski & Günnemann (2019); D.Zugner et al. (2019); D. Zügner & Günnemann (2018); Akbarnejad & Günnemann (2019) adopt edge-flipping and node attribute perturbation to poison graph data mining pipelines, e.g., graph neural networks and graph embedding models. The key idea is to introduce relaxed surrogate functions to the combinatorial attack objective and then solve the relaxed optimization problem instead. Notably, D.Zugner et al. (2019); D. Zügner & Günnemann (2018); Akbarnejad & Günnemann (2019) define the attack objective as a sum of smallest eigenvalues of the adjacency matrix of a given graph. Though it is not explicitly claimed, it is intrinsically a submodular maximization problem.

Qi et al. (2019); Wang et al. (2020) unveil that simple greedy search can deliver provably effective attacks against DNN classifiers without domain-specific heuristics, if all the link weights between neurons are non-negative. Non-negativity of the parameter guarantees strict submodularity of the attack objective. However, it brings significantly deterioration of the classifier's accuracy, which is not applicable for real-world learning tasks.

# 3 ROBUSTNESS CHARACTERIZATION AND ASSESSMENT

We assume an input instance $x = \{x_1, x_2, x_3, ..., x_n\}$ of $n$ categorical attributes. Each $x_i$ takes any of $m$ ($m \geq 1$) categorical values. The classifier $f$ outputs decision probabilities $f_{y_k}$ ($k = 1,2,3,...,K$) with respect to different class labels. In practices, each categorical value of $x_i^j$ is cast to a $D$-dimensional pre-trained embedding vector, e.g., $e_i^j \in \mathbb{R}^D, j = 1,2,...,m$. To represent an instance $x$ with the embedding vectors of its category values, we define binary variables $b = \{b_i^j\}$, $i=1,2,...,n$, $j=1,2,...,m$, where $b_i^j = 1$ when the $j$-th attribute value is present for $x_i$ and $b_i^j = 0$ otherwise. An instance $x$ can then be represented by an $\mathbb{R}^{n*m*D}$ tensor with $x_{\{i,j,:\}} = b_i^j e_i^j$. Let $\hat{b} = \{\hat{b}_i^j\}$ indicate the adversarial modifications introduced into $b$. For a perturbed $\hat{x}$, its $\hat{b} \neq b$. Depending on the type of attacks to implement, e.g., *insertion*, *deletion* or *substitution*, $\hat{b}$ differs from $b$ in different ways. Without loss of generality, let $y_K$ denote the true class label of $x$ and all the other $y_k$ ($k = \{1,...,K-1\}$) are the potential targets of an evasion attack. The goal of attack is to make $f_{y_K}(x,\hat{b})$ as low as possible and increase $f_{y_k}(x,\hat{b})$ of a fixed $k$ (*targeted attack*) or any $k$ of $\{1,...,K-1\}$ (*non-targeted attack*) as high as possible simultaneously. We focus on *non-targeted evasion attack*, and leave the targeted scenario for future study.

## 3.1 INFORMATION-THEORETIC CHARACTERIZATION OF ADVERSARIAL VULNERABILITY

**Theorem 1** *For an instance $(x,y)$ and a training set $S$ sampled from the same underlying distribution $\mu_{x,y}$, $f \in \mathcal{H}$ is trained using $S$ with a deterministic training paradigm. The expected adversarial risk $\mathcal{R}_\varepsilon^{adv}$ defined in Eq.1 can be bounded as in Eq.2, if the loss function $\ell$ in $\mathcal{R}_\varepsilon^{adv}$ adopts the zero-one loss.*

$$\mathcal{R}_\varepsilon^{adv} \geq 1 - \frac{2I(x;y,S) - 2\eta_x - I(f_y;S)/2 + const}{\log K} \tag{2}$$

$$\eta_x = \sup_{|diff(b,\hat{b})| \leq \varepsilon} I(x;y,S) - I(\hat{x};y,S) \tag{3}$$

*where const denotes a constant term. $diff(b,\hat{b})$ indicates the set of the categorical attributes modified in the attack. $I(x;y,S)$ denotes the mutual information between the feature $x$ and the pair of label $y$ and training set $S$. $\eta_x$ is the supremum of the difference between the mutual information before and after adversarial perturbation, noted by $I(x;y,S)$ and $I(\hat{x};y,S)$ respectively. $I(f_y;S)$ denotes the mutual information between $S$ and $f$.*

Theorem 1 unveils the three impact factors jointly determining adversarial vulnerability of the targeted classifier $f$ (for answering Q1 in Section 1). The proof is given in Appendix.A. *In Eq.2, a lower mutual information $I(x;y,S)$ indicates higher adversarial risk*, i.e., $\mathcal{R}_\varepsilon^{adv}$ has a higher lower bound. A lower

$I(x;y,S)$ denotes weaker consistency between the input instance $(x,y)$ and the training set $S$. The classifier trained by $S$ thus produces weaker decision confidence for such $(x,y)$. With $(x,y)$ dropped to the ambiguous zone near the classification boundary, the classifier is more prone to adversarial perturbation.

*A higher $I(f_y;S)$ leads to a higher adversarial risk according to Eq.2.* The mutual information $I(f_y;S)$ reflects the dependence between the classifier $f$ and the training set $S$, which can be considered as a lower bound of the VC-dimension for countable hypothesis space $f \in \mathcal{H}$ (Xu & Raginsky, 2017; Zhu et al., 2020). A higher $I(f_y;S)$ thus denotes that $f$ is more likely to suffer from being overfitted to $S$. Model overfiting is one of the causes of adversarial vulnerability (Tu et al., 2019). Resonating with the association between $I(f_y;S)$ and adversarial risk, three popularly used robustness enhancement methods controlling the classifier's complexity and overfiting risk can potentially adjust the adversarial vulnerability of the classifier with categorical inputs. **1) Adversarial training** (Miyato et al., 2016; Sinha et al., 2018; Florian et al., 2018; Wang et al., 2019; Shafahi et al., 2019). The adversarially retrained classifier $\hat{f}$ is less correlated with the original training set $S$ compared to $f$, which reduces $I(f_y;S)$ and may help mitigate the adversarial threat. **2)** The addition of **nuclear norm regularization** over the classifier's parameters in the training process (Ravi et al., 2019; Tu et al., 2019). The resultant regularized classifier has a controlled model complexity, which can potentially reduce the adversarial risk. **3) Random smoothing** (Lee et al., 2019; Levine & Feizi, 2020; Dvijotham et al., 2020; Boj. et al., 2020). Following (Cohen et al., 2019), this defense method randomly selects and flips the input categorical features of the targeted classifier. The randomly perturbed classification output is less correlated with the training data $S$'s distribution, which may reduce adversarial risk.

*A higher $\eta_x$ in Eq.3 indicates a higher loss of predicative information by changing $x$ to $\hat{x}$, and thus a higher adversarial risk.* In this sense, the vulnerability of the classifier depends on the sensitivity of the attacked categorical features with respect to the classification task.

### 3.2 Scoring Robustness via Set Function Optimization

We formulate the assessment of adversarial robustness level on categorical data as *a set function maximization problem*: we aim at finding a minimal set of categorical feature perturbations $l = diff(b,\hat{b})$, with which the maximum gap between the classification confidence on any wrong label $k$ and the correct label $K$ is larger than a predefined threshold $\Gamma$. The size of the minimal perturbation set $l$ is used to assessment adversarial robustness of $f$ over the input instance. Within the perturbation budget $\varepsilon$, a smaller/larger $|l|$ provoking the alert indicates that the classifier's output on the input $x$ is more/less vulnerable w.r.t. adversarial perturbation in the categorical feature space. We consider both the pessimistic (in Eq. 4) and optimistic scenario (in Eq. 5) from the defender's perspective for robustness scoring.

$$\psi^{pess}(l) = \max_{\hat{b}_{i=1\ldots n}, l=diff(b,\hat{b})} \quad \max_{\hat{b}_i^{j=1\ldots m}} (m_f) \qquad s.t. \quad m_f \geq \Gamma, |l| \leq \varepsilon \qquad (4)$$

$$\psi^{optim}(l) = \min_{\hat{b}_{i=1\ldots n}, l=diff(b,\hat{b})} \quad \max_{\hat{b}_i^{j=1\ldots m}} (m_f) \qquad s.t. \quad m_f \geq \Gamma, |l| \leq \varepsilon \qquad (5)$$

where $m_f = \max_{k \in \{1,\ldots,K-1\}} \{f_{y_k}(x,\hat{b})\} - f_{y_K}(x,\hat{b})$ is the gap defined with an input instance $x$. $m_f \in [-1,0)$ and $m_f \geq 0$ correspond to correct and wrong classification respectively. $\Gamma$ is a threshold reflecting different levels of tolerance to the perturbation of the classifier's output. $m_f \geq \Gamma$ then triggers an alert of potential adversarial attack. Safety-critical domains, such as cyber security, prefer a lower $\Gamma$ to provoke alerts even with small yet suspicious input perturbations.

**Observation 1** *With the threshold $\Gamma = 0$, $\psi^{pess} \geq 0$ derived by solving the pessimistic assessment problem in Eq.4 produces an empirical estimate of the expected adversarial risk $f$ with an input instance $(x,y)$ sampled from $\mu_{x,y}$.*

We leave the derivation of this observation in Appendix.B. Eq. 4 is a bi-level optimization problem. The inner level seeks the optimal change of the category value of a given categorical feature, in order to produce maximum possible perturbation on the classifier's output. The outer level finds the optimal combination of different categorical features for maximizing $m_f$ within the attack budget. $|l|$ is thus a **pessimistic** robustness assessment with respect to **a knowledgeable adversary**. The adversary can locate the most effective modification for both individual categorical features and the combination set of the categorical features.

In contrast, $|l|$ to Eq. 5 reflects an **optimistic** robustness assessment with respect to **an oblivious adversary**. In practices, an adversary can be limited by the knowledge about the targeted classifier

and/or the query budget. Such an oblivious adversary can only locate the effective category values to attack for a given categorical feature, while can't find the optimal feature combinations to solve the outer-level optimization.

## 3.3 Provably Accurate Robustness Assessment via Weakly Submodular Maximization

Exactly solving Eq.4 and Eq.5 is still NP-hard. To answer Q2 in Section 1, Theorem 2 and Theorem 3 elaborate the optimality guarantee of solving both robustness assessment problems via computationally efficient greedy search, for any classifier meeting the smoothness condition in Definition.2.

**Definition 2** *Smoothness Condition of* $f$. *Let* $\Omega = (p,q)$, $p,q \in \mathbb{R}^n$ *and* $f: \mathbb{R}^n \to \mathbb{R}$ *be a Lipschitz-continuous and differentiable function. A function* $f$ *is* $(m_\Omega, M_\Omega)$-*smooth on* $\Omega$, *if for any* $(p,q) \in \Omega$, $m_\Omega \in \mathbb{R}$ *and* $M_\Omega \in \mathbb{R}^+$, $\epsilon = f(q) - f(p) - \langle \nabla f(p), q-p \rangle$ *satisfies:*

$$\frac{m_\Omega}{2} \|q-p\|_2^2 \le |\epsilon| \le \frac{M_\Omega}{2} \|q-p\|_2^2. \tag{6}$$

Different from (Elenberg et al., 2018), Eq.6 controls only the fluctuation of the function value by varying the input from $p$ to $q$ and vice versa, regardless of convexity or concavity of $f$. It allows broader choices of the classifier's architecture. With $\gamma$-weak submodularity (Elenberg et al., 2018; Santiago & Yoshida, 2020) defined in Appendix C, we establish Theorem 2.

**Theorem 2** *Let* $\Omega_\zeta = \{(\hat{b}, \hat{b}') : |diff(b,\hat{b})| \le \zeta, |diff(b,\hat{b}')| \le \zeta, |diff(\hat{b}, \hat{b}')| \le \zeta, \zeta \ge 1\}$, *where* $\hat{b}$ *and* $\hat{b}'$ *denote two sets of attribute changes. If the classifier* $f_{y_k}(x)$ $(k=1,...,K)$ *follows the regularity condition given by* $(m_{k,\Omega_\zeta}, M_{k,\Omega_\zeta})$-*smoothness constraint on* $\Omega_\zeta$, *the pessimistic and optimistic robustness assessment (with Eq.4 and Eq.5) can be formulated respectively as monotone and non-monotone* $\gamma_\zeta$-*weakly submodular maximization. Let* $\epsilon_k = f_{y_k}(x,\hat{b}') - f_{y_k}(x,\hat{b}) - \langle \nabla f_{y_k}(x,\hat{b}), \hat{b}' - \hat{b} \rangle$, *and* $\nabla f_y(x,b)_\nu$ *denote the elements of* $\nabla f_y(x,b)$ *corresponding to the difference between* $\hat{b}$ *and* $\hat{b}'$, *where* $\nu = diff(\hat{b}, \hat{b}')$. *The submodularity ratio* $\gamma_\zeta^{pess}$ *for the pessimistic robustness assessment is bounded as:*

$$\gamma_\zeta^{pess} = \min_{k=1,...,K} \{\gamma_{k,\zeta}^{pess}\} \tag{7}$$

*where* $\gamma_{k<K,\zeta}^{pess} \ge \frac{\|\nabla f_{y_k}(x,b)_\nu\|_2 + m_{k,\Omega_1}|\zeta|/2}{\|\nabla f_{y_k}(x,b)_\nu\|_2 + M_{k,\Omega_\zeta}|\zeta|/2}$ *for* $\epsilon_k \ge 0$, $\gamma_{k<K,\zeta}^{pess} \ge \frac{2m_{k,\Omega_\zeta}}{\|\nabla f_{y_k}(x,b)_\nu\|_2^2}(\|\nabla f_{y_k}(x,b)_\nu\|_2 - M_{k,\Omega_1}|\zeta|/2)$ *for* $\epsilon_k < 0$, $\gamma_{K,\zeta}^{pess} \ge \frac{2m_{K,\Omega_\zeta}}{\|\nabla f_{y_K}(x,b)_\nu\|_2^2}(\|\nabla f_{y=K}(x,b)_\nu\|_2 - M_{K,\Omega_1}|\zeta|/2)$ *for* $\epsilon_K \ge 0$, *and* $\gamma_{K,\zeta}^{pess} \ge \frac{\|\nabla f_{y_K}(x,b)_\nu\|_2 + m_{K,\Omega_1}|\zeta|/2}{\|\nabla f_{y_K}(x,b)_\nu\|_2 + M_{K,\Omega_\zeta}|\zeta|/2}$ *for* $\epsilon_K < 0$. *Similarly, the submodularity ratio* $\gamma_\zeta^{optim}$ *of the optimistic assessment is:*

$$\gamma_\zeta^{optim} = \min_{k=1,...,K} \{\gamma_{k,\zeta}^{optim}\} \tag{8}$$

*where* $\gamma_{k<K,\zeta}^{optim} \le \frac{\|\nabla f_{y_k}(x,b)_\nu\|_2^2}{2M_{k,\Omega_\zeta}(\|\nabla f_{y_k}(x,b)_\nu\|_2 - m_{k,\Omega_1}|\zeta|/2)}$ *for* $\epsilon_k \ge 0$, $\gamma_{k<K,\zeta}^{optim} \le \frac{\|\nabla f_{y_k}(x,b)_\nu\|_2 + m_{k,\Omega_\zeta}|\zeta|/2}{\|\nabla f_{y_k}(x,b)_\nu\|_2 + M_{k,\Omega_1}|\zeta|/2}$ *for* $\epsilon_k < 0$, $\gamma_{K,\zeta}^{optim} \le \frac{\|\nabla f_{y_K}(x,b)_\nu\|_2 + m_{K,\Omega_\zeta}|\zeta|/2}{\|\nabla f_{y_K}(x,b)_\nu\|_2 + M_{K,\Omega_1}|\zeta|/2}$ *for* $\epsilon_K \ge 0$, *and* $\gamma_{K,\zeta}^{optim} \le \frac{\|\nabla f_{y_K}(x,b)_\nu\|_2^2}{2M_{K,\Omega_\zeta}(\|\nabla f_{y_K}(x,b)_\nu\|_2 - m_{K,\Omega_1}|\zeta|/2)}$ *for* $\epsilon_K < 0$.

Enjoying the weak submodularity, the pessimistic and optimistic assessment can be derived by greedy search with provable accuracy. We present a *Forward Stepwise Greedy Search* (FSGS) algorithm for deriving the pessimistic assessment (Eq.4). Due to the non-monotone objective function of the optimistic assessment (Eq.5), a stochastic variant of the standard *FSGS*, namely *Randomized FSGS* (RandGS) is adopted, following the suggestion in (Buchbinder et al., 2014). We give the pseudo codes of *FSGS* and *RandGS* in Algorithm.1 and 2 in Appendix.D. Compared to *FSGS*, the difference in *RandGS* is to **randomly select one of the top-ranked candidate features** contributing with the largest marginal gain $m_f$ to the objective function. Theorem 3 gives the approximation guarantee of the pessimistic and optimistic robustness assessment using *FSGS* and *RandGS* respectively.

**Theorem 3** *For a* $(m_{\Omega_\zeta}, M_{\Omega_\zeta})$-*smooth classifier* $f$, *the approximation quality of the solution* $l$ *to Eq.4 and Eq.5 is bounded respectively as*

$$|m_f(l)| + e^{-\gamma_\zeta^{pess}} \le (1 - e^{-\gamma_\zeta^{pess}})|m_f(l_{pess}^*)| \quad \text{(Pessimistic Assessment)}$$
$$|m_f(l)| \ge \frac{1}{\gamma_\zeta^{optim}} e^{-\gamma_\zeta^{optim}} |m_f(l_{optim}^*)| \quad \text{(Optimistic Assessment)} \tag{9}$$

*where* $\gamma_\zeta^{pess}$ *and* $\gamma_\zeta^{optim}$ *are the submodularity ratio in Eq. 7 and Eq. 8, respectively.* $l_{pess}^*$ *and* $l_{optim}^*$ *are the true optimal solutions to the two problems.*

With Theorem 3, the Q2 in Section.1 is fully addressed. Notably, any classifier with a finite Lipshitz constant meets the smoothness condition. For practical estimation, $M_{\Omega_\zeta}$ can be computed as local Lipshitz constant of $f$ if $\epsilon > 0$ in Definition.2 (or $-f(x)$ if $\epsilon < 0$). Furthermore, $m_{\Omega_\zeta}$ can be calculated as the local strong convexity constant of $f(x)$ if $\epsilon > 0$ in Definition.2 (or $-f(x)$ if $\epsilon < 0$). Recent works (Fazlyab et al., 2019; Jordan & Dimakis, 2021) have shown that a large number of neural networks are Lipschitz continuous, such as the fully-connected neural networks with *ReLU*, *sigmoid*, and *tanh* activation functions. The *FSGS* based robustness assessment method can thus be applied to most DNN-based classifiers that are popularly adopted in real-world applications.

Theorem 3 also reveals that the role that *the smoothness of $f_y$ plays in characterizing the solvability of the robustness assessment problem*. As indicated by Eq. 7 and Eq. 8, a smoother $f_k$ with smaller $m_{k,\Omega_\zeta}$ and $M_{k,\Omega_\zeta}$ has a higher submodularity ratio ($\gamma_{k,\zeta}^{pess}$ and $\gamma_{k,\zeta}^{optim}$). With a submodularity ratio closer to 1, both *FSGS* and *RandGS* can obtain better approximation quality to the underlying optimal result of the robustness assessment, according to Eq. 9. Comparing Eq. 7 to 8, the submodularity ratio of the pessimistic assessment is higher than that of the optimistic opponent. The greedy search has lower approximation quality under the optimistic setting. The optimistic assessment is thus generally higher than the pessimistic assessment. It echos with the intuition that a classifier is more vulnerable to a knowledgeable adversary than an oblivious one.

### 3.4 ORTHOGONAL MATCHING PURSUIT GREEDY SEARCH FOR ROBUSTNESS ASSESSMENT

Both *FSGS* and *RandGS* need to traverse all the combinations of candidate attributes and the subsets of the selected set of attributes. In the worst case, they require $\sum_{t=0}^{T}((n-t)*m*2^t)$ objective function evaluations for $T$ iterations, where $n$ denotes of the number of categorical features in an instance $x$ and each feature can choose any of the $m$ categorical values. Large $n$ and $m$ are usually witnessed in the real-world data. Both methods thus are costly in querying the oracle classifier in such real-world cases.

In (Wang et al., 2020), an orthogonal matching pursuit guided greedy searching strategy (*OMPGS*) is introduced to address the query bottleneck of primary greedy search in evasion attack on discrete data. In the context of attack, this method computes $\nabla f_y(x)_{\hat{b}}$ (the gradient of $f_y$ w.r.t. the binary indicators $\hat{b}$ of unmodified categorical features). The magnitude of the gradient measures the capability of each candidate feature in reducing the classification confidence of the correct label. In this sense, *OMPGS* narrows down the candidate attributes to those that best correlate with the orthogonal complement of the subset of the attributes already selected in greedy search. The query complexity of *OMPGS* is thus independent of $n$ and $m$. Due to the space limit, the algorithm implementation of *OMPGS* are explained in Algorithm.3 in Appendix.E.

We apply *OMPGS* to speed up the solutions to both the pessimistic and optimistic assessment problems. The approximation guarantee of *OMPGS* given in (Wang et al., 2020) only explains the provably accuracy of using *OMPGS* in *the pessimistic assessment*. In this work, we further bound the approximation accuracy of *OMPGS* to derive *the optimistic assessment* in Theorem 4. Comparing to Theorem 3, *OMPGS* achieves a lower but similar approximation accuracy to the underlying optimal solution to the optimistic assessment (Eq. 5).

**Theorem 4** *Following the notations in Theorem 2 and Theorem.3, the approximation quality of the optimistic assessment score $|l|$ obtained by OMPGS is bounded as in the followings:*

$$|m_f(l)| \geq \frac{1}{\psi}e^{-\psi}|m_f(l^*_{optim})|, \ \psi = \max\{\varphi_k\}(k=1,2,3,...,K)$$

$$\varphi_k = \max\{\frac{\|\nabla f_{y_k}(x,b)_\nu\|_2^2}{2M_{k,\Omega_\zeta}(\|\nabla f_{y_k}(x,b)_\nu\|_2 - m_{k,\Omega_1}|\zeta|/2)}, \frac{\|\nabla f_{y_k}(x,b)_\nu\|_2 + m_{k,\Omega_\zeta}|\zeta|/2}{\|\nabla f_{y_k}(x,b)_\nu\|_2 + M_{k,\Omega_1}|\zeta|/2}\}$$

(10)

In both of the pessimistic and optimistic assessment, the gradient magnitude used by *OMPGS* can be interpreted as an upper bound of the increase of $m_f$ by adding adversarial perturbation to a previously unmodified feature, as shown in Appendix E. Notably, the gradient used in *OMPGS* is not taken directly with respect to the categorical features. Instead, the gradient is taken with respect to the binary indicator variable $b_i^j$ (see Section.3). We rank the categorical features according to the magnitudes of the gradients with respect to different $b_i^j$. A higher gradient magnitude indicates that the corresponding categorical feature $x_i = x_i^j$ can contribute higher marginal gain of $m_f$. *OMPGS* narrows down range of greedy search. Only the top ranked candidate features are considered to modify.

Table 1: Pessimistic and Optimistic Robustness Assessment $|l|$. A lower/higher assessment score indicates weaker/stronger robustness level. SR is the success rate of attack. OVF indicates SR=0.

| GreedySeearch | P/O | $\Gamma$ | Yelp | | IPS | | Splice | |
| --- | --- | --- | --- | --- | --- | --- | --- | --- |
| | | | SR | Med (Avg) | SR | Med(Avg) | SR | Med(Avg) |
| **FSGS/ RandGS** | P | -0.4 | 0.98 | 1 (1.5) | 0.80 | 1 (1.2) | 0.91 | 1 (1.7) |
| | | 0 | 0.98 | 1 (1.9) | 0.80 | 1 (1.2) | 0.91 | 1 (1.7) |
| | O | -0.4 | 0.09 | 9 (8.1) | 0.33 | 1 (1.2) | 0 | OVF |
| | | 0 | 0.09 | 12 (11.8) | 0.33 | 1 (1.3) | 0 | OVF |
| **GradAttack** | P | -0.4 | 0.84 | 1 (1.5) | 0.59 | 1 (1.8) | 0.72 | 3 (3.7) |
| | | 0 | 0.76 | 2 (1.7) | 0.59 | 1 (1.7) | 0.72 | 3 (3.7) |
| **OMPGS/ RandOMPGS** | P | -0.4 | 0.98 | 1 (2.0) | 0.91 | 1 (1.6) | 0.74 | 2 (2.1) |
| | | 0 | 0.97 | 2 (2.5) | 0.91 | 1 (1.8) | 0.74 | 2 (2.3) |
| | O | -0.4 | 0.63 | 7 (6.9) | 0.35 | 3 (3.4) | 0 | OVF |
| | | 0 | 0.52 | 8 (7.6) | 0.35 | 3 (3.6) | 0 | OVF |

## 4 EXPERIMENTAL STUDY

We instantiate the study with standard LSTM based classifiers trained on three multi-class datasets collected from real-world applications of *Text analysis*, *Cyber security* and *Biomedicine*.

**Yelp-5 (Yelp)**(Asghar, 2016). The Yelp-5 reviews dataset was obtained from the Yelp Dataset Challenge in 2015. We use the reviews containing 650K training and 50K testing samples with the classes from 1 star to 5 stars for learning classifier. Each word is a categorical feature with 300-dim embedding.

**Intrusion Prevention System Dataset (IPS)** (Wang et al., 2020). The IPS dataset has 242,467 instances, each of which is a 20-step attack sequence and represented by $x \in \mathbb{R}^{20*1103*70}$, because each step can be a categorical value from 1,103 different malicious actions embedded in 70-dim space by word embedding. An $x$ is classified with 3 labels: *attack type 1, attack type 2*, and *others*.

**Splice-junction Gene Sequences (Splice)** (Noordewier et al., 1991). This dataset has 3,190 samples. Each one is a gene fragment of 60 categorical features with values in {'A', 'G', 'C', 'T', 'N'}. Learning an embedding for each value with 30-dim, one instance is given as $x \in \mathbb{R}^{60*5*30}$. The task is to predict each splice instance as an intro-exon boundary (*IE*), an exon-intro boundary (*EI*), or neither of them.

The LSTM-based classifiers with *ReLu* activation function and dropout achieve **accuracy** scores of *0.61*, *0.92* and *0.95* respectively for *Yelp*, *IPS* and *Splice*. More details about the datasets and the experimental setting are given in Appendix F.1. To strengthen the universality of the robustness assessment framework, we also include additional results on binary-classification datasets using Convolution Neural Networks (CNN) in Appendix.F.5

The experimental study is to address questions below:

**1)** What's the robustness assessment result of a targeted classifier? (Empirically evaluate the answer in Theorem 2,3 and 4 to Q2 in Section 4.1). For the classifier trained on each dataset, we first find the robustness assessment $|l|$ for each data instance. We report the median and mean value of all the $|l|$ derived on the dataset. A lower/higher median value indicates a weaker/stronger robustness of the classifier to the adversarial attack over the given data set. The tolerance threshold $\Gamma$ is tested on $-0.4$ and 0 to assess our proposed assessment method with varied tolerance to adversarial threats in safety-critical applications. Different greedy search methods are compared on their assessment result. Besides *FSGS* and *OMPGS* (for solving Eq.4) and *RandGS* and *RandOMPGS* (for solving Eq.5) , we also include *Gradient-based Attack (GradAttack)* from (Qi et al., 2019) in the evaluation. *GradAttack* tackles only the pessimistic assessment problem. The comparison with *GradAttack* is to verify the merit of *FSGS* and its variants in solving the robustness assessment problem of a general classifier. For *RandGS* and *OMPGS*, we empirically choose 10 candidate features in each iteration of greedy search[1].

**2)** Can the information-theoretic characterization of adversarial vulnerability in Theorem 1 be verified using the proposed robustness assessment? (Empirically evaluate the answer in Theorem 1 to Q1 in Section 4.2 and 4.3). We verify an increase of risk on $x_{l_{mi}}$ with lower mutual information, i.e., $I(x_{l_{mi}};y,S) < I(x;y,S)$, and verify an increase of robustness by reducing $I(f_y;S)$.

**3)** We compare **the time complexity** of *FSGS*,*OMPGS* and *GradAttack*. Furthermore, we evaluate **the averaged running time** and **the averaged iterations** of the three methods over *Yelp*, *IPS* and *Splice* datasets. As shown, both *FSGS* and *OMPGS* stop early the search with less loops than *GradAttack*. *OMPGS* costs much fewer queries and less running time than *FSGS*. Compared to *GradAtttack*, *OMPGS* has a better chance to select the really useful feature modifications, thus costing fewer loops. We provide the results in Appendix.F.3 due to the space limit.

---

[1]Implementations are available at `https://github.com/XYZ211923Y/-RobustXXXXX`.

## 4.1 ROBUSTNESS ASSESSMENT WITH GREEDY SEARCH

Besides showing the median and mean of the assessment score $|l|$ (*Med(Avg)*) of the testing instances, we also report attack success rate (noted as **SR** in the tables), i.e., the fraction of the testing instances on which the greedy methods can push $m_f$ greater than the tolerance threshold within the one-hour time limit. With similar median/mean values of $|l|$, a lower **SR** also indicates stronger robustness of the classifier. If the greedy search fails to push $m_f$ to surpass the threshold $\Gamma$ for all the testing instances within the time limit, we note this situation as *OVF* (an abbreviation for overflow) and **SR** = 0 in the tables. $P/O$ denotes the *pessimistic* or *optimistic* assessment respectively.

Table 1 shows several interesting findings. First, for both the pessimistic and optimistic robustness assessment, **a lower tolerance threshold $\Gamma$ results in lower scores over all the datasets**. This observation is consistent with the intuition of setting the tolerance threshold. A lower tolerance threshold helps choose more worst-case resilient classifiers. Our study allows for the analysis with different vigilance levels. Second, as expected, **the pessimistic robustness assessment scores are always significantly lower than those of the optimistic assessment, and the $SR$ value is much lower on the optimistic assessment scenario.** Due to lack of query capability and knowledge about $f$, it is difficult for the oblivious adversary to locate the sensitive features.

*OMPGS* and *RandOMPGS* produce similar robustness assessment scores, compared to *FSGS* and *RandGS*. Compared to evaluate each candidate features as in *FSGS/RandGS*, *OMPGS* improves the search efficiency using the gradient based heuristic guidance. On *IPS* and *Splice*, *FSGS/RandGS* needs 217.65 and 1123.51 seconds on average on each data sample for the pessimistic and optimistic assessment. In contrast, *OMPGS/RandOMPGS* costs only 0.24 and 1.59 seconds for the pessimistic and optimistic assessment, more than 700 times faster than *FSGS/RandGS*. *GradAttack*, in contrast, produces much lower *SR* than *FSGS* and *OMPGS* in the pessimistic assessment setting. As discussed in Section.2, the greedy search in *GradAttack* ignores the combinatorial effect of the candidate features and the previously modified features, which leads to worse approximation quality and much less efficient attacks compared to *FSGS* and *OMPGS*. The results echo the theoretical analysis. With significantly lower *SR*, *GradAttack* can not provide an accurate answer to the robustness assessment problem.

We conduct one-factor-at-a-time sensitivity analysis on each dataset (Campbell et al., 2008) to measure the sensitivity of features in the classification task. Given a data instance, we change each feature while keeping all the others fixed. The averaged change of the classifier's output over all the testing instances is used as the feature-wise sensitivity measurement. A larger value indicates that the classifier's output is more sensitive to the change over the corresponding feature. We find that the top-sensitive features also appear as the most frequently selected features by *FSGS*,*GradAttack* and *OMPGS*. Details about the sensitivity analysis can be found in Appendix.F.4. The interesting overlapping between the attributes which are useful for attack and the top-ranked sensitive attributes confirms our intuition about the association between feature sensitivity and adversarial vulnerability of the classifier.

## 4.2 ROBUSTNESS VS FEATURE INFORMATIVENESS

According to the discussion in Theorem 1, a lower $I(x;y,S)$ indicates a higher adversarial risk on the input instance $(x,y)$, implying a smaller robustness assessment score. To verify such impact of feature informativeness, we compare the robustness on $x$ and on $x_{l_{mi}}$ that has a lower mutual information, i.e., $I(x_{l_{mi}};y,S) < I(x;y,S)$. The sample $x_{l_{mi}}$ is constructed as those on the half-way of pushing the $m_f$ of the original sample $x$ to surpass $\Gamma$. For each dataset, we select the instances requiring at the least two modified features in order to deliver a successful attack. These instances are treated as *original*. There are 125, 414 and 324 *original* instances in *Yelp*, *IPS*, and *Splice* respectively. The results in Table 2 confirm that the classifier becomes less robust to adversarial modifications over $x_{l_{mi}}$ samples in the ambiguous zone, i.e., higher SR and smaller $|l|$ on the $x_{l_{mi}}$ samples than the original samples $x$.

## 4.3 ROBUSTNESS WITH DEFENSIVE TECHNIQUES

As discussed in Theorem.1, reducing $I(f_y;S)$, the dependence of a classifier over the training set $S$, improves the classifier's adversarial robustness. We tune a classifier via *adversarial training* (**AdvC**), *the nuclear-norm based regularization* (**NuR**) following (Ravi et al., 2019) and *random smoothing* by (**RS**) as suggested in (Boj. et al., 2020) to reduce $I(f_y;S)$, and then evaluate if the pessimistic robustness assessment becomes higher, e..g, decrese of SR. To adopt adversarial training, we solve Eq. 4 using

Table 2: Robustness Assessment vs Feature Informativeness, reported at $\Gamma=0$. $I(x_{l_{mi}};y,S)<I(x;y,S)$ results in higher adversarial risk (higher SR, smaller $|l|$) on $x_{l_{mi}}$ than original $x$.

| GreedySeearch | P/O | Sample | Yelp | | IPS | | Splice | |
|---|---|---|---|---|---|---|---|---|
| | | | SR | Med(Avg) | SR | Med(Avg) | SR | Med(Avg) |
| FSGS/ RandGS | P | original $x$ | 0.98 | 1 (1.9) | 0.80 | 1 (1.2) | 0.91 | 1 (1.7) |
| | | $x_{l_{mi}}$ | 1.00 | 1 (1.7) | 0.95 | 1 (1.1) | 0.99 | 1 (1.2) |
| | O | original $x$ | 0.09 | 12 (11.8) | 0.33 | 1 (1.3) | 0 | OVF |
| | | $x_{l_{mi}}$ | 0.94 | 6 (6.0) | 0.65 | 1 (1.2) | 0 | OVF |
| OMPGS/ RandOMPGS | P | original $x$ | 0.97 | 2 (2.5) | 0.91 | 1 (1.8) | 0.74 | 2 (2.3) |
| | | $x_{l_{mi}}$ | 0.99 | 2 (2.4) | 0.93 | 1 (1.3) | 0.94 | 1 (1.5) |
| | O | original $x$ | 0.97 | 4 (4.6) | 0.35 | 3 (3.6) | 0 | OVF |
| | | $x_{l_{mi}}$ | 1.00 | 4 (4.5) | 0.57 | 3 (3.3) | 0 | OVF |

Table 3: Pessimistic Robustness Assessment ($\Gamma = 0$) after Robustness Enhancement by reducing $I(f_y;S)$. $\downarrow^{SR}$ is the the amount of SR decreasing comparing to the SR value with $\Gamma=0$ in Table 1.

| Model | Algo. | Yelp | | | IPS | | | Splice | | |
|---|---|---|---|---|---|---|---|---|---|---|
| | | SR | $\downarrow^{SR}$ | Med(Avg) | SR | $\downarrow^{SR}$ | Med(Avg) | SR | $\downarrow^{SR}$ | Med(Avg) |
| AdvC | FSGS | 0.84 | 0.13 | 1 (1.6) | 0.74 | 0.06 | 1 (1.4) | 0.85 | 0.06 | 2 (2.1) |
| | OMPGS | 0.83 | 0.15 | 2 (2.5) | 0.60 | 0.31 | 2 (2.2) | 0.65 | 0.09 | 2 (2.8) |
| NuR | FSGS | 0.72 | 0.25 | 1 (1.9) | 0.79 | 0.01 | 1 (1.1) | 1 | -0.09 | 1 (1.4) |
| | OMPGS | 0.68 | 0.30 | 2 (2.4) | 0.63 | 0.28 | 1 (1.0) | 0.72 | 0.02 | 1 (2.5) |
| RS | FSGS | 0.76 | 0.21 | 1 (1.8) | 1.00 | -0.20 | 1 (1.4) | 0.91 | 0 | 1 (1.9) |
| | OMPGS | 0.74 | 0.24 | 2 (2.2) | 0.95 | -0.04 | 1 (1.7) | 0.78 | -0.04 | 2 (2.2) |
| SRS (Boj. et al., 2020) | | 1.0 | - | 2 | 1.0 | - | 2 | 1.0 | - | 1 |

*OMPGS* until the value of $m_f$ drops into the interval $[-0.2,0.0]$. The resultant modified instances are used for adversarial training. The regularization parameter of the nuclear-norm regularized training method is chosen following (Ravi et al., 2019). The random smoothing is conducted by randomly flipping the values of input categorical features (**RS**) with $p_- = 0.6$ and $p_+ = 0.01$ as suggested in (Boj. et al., 2020). We also implement the original sparsity-aware random smoothing based certifiable robustness (*SRS*) (Boj. et al., 2020) as a reference method of random smoothing based defense techniques. It performs first the same randomly flipping of categorical features as in **RS**. The least number of the features changed to trigger a wrong classification, a.k.a. *certifiable robustness*, is then estimated via a dimension-independent evaluation based on the randomly flipped features. **SRS** (Boj. et al., 2020) is reported to be significantly more efficient and produce tighter robustness assessments than other random smoothing-based methods (Lee et al., 2019; Levine & Feizi, 2020; Dvijotham et al., 2020).

Table 3 illustrates the pessimistic robustness assessment scores of the tuned classifiers with the tolerance threshold $\Gamma=0$. We skip the results of the optimistic robustness assessment and the results with $\Gamma=-0.4$ as they present a similar variation tendency. Both adversarial training and nuclear-norm regularization help improve the classifier's robustness, leading to lower *SR* and higher $|l|$ over all the three datasets in most of the cases. However, on *Splice*, the adversarial vulnerability reduction effect is much less visible than those observed on the other two datasets. We ascribe this phenomenon to the fact that the feature-wise sensitivity on *Splice* has a more skewed distribution than the other datasets. A few categorical features in *Splice* are significantly more sensitive than the others in the classification task, corresponding to important proteins dominating the molecules' properties (see feature sensitivity analysis in Section 4.1 and Appendix.F.4). The existence of such highly sensitive features makes it difficult to enhance the targeted classifier's robustness without losing the classification utility. *RS* and *SRS* have higher SR than the other robustness enhanced models, showing that the random smoothing technique helps the least improve robustness or even make it more vulnerable to the attack, in the case where a fraction of categorical features are highly sensitive. As seen in Table.1, highly sensitive features can be witnessed in many diversified applications. Firstly, flipping these sensitive features can lead to miss-classification rather than providing robustness. Secondly, if random flipping misses the sensitive features, the adversary can still target at these features to deliver a successful attack.

## 5 CONCLUSION

We unveil the key factors that jointly determine the adversarial vulnerability of a classifier with categorical inputs via information-theoretic analysis. The characteristics of the categorical data and the functional property of the targeted classifier act as the external and internal cause of the adversarial threat. We further set a theoretical association between the smoothness of the targeted classifier and the solvability of both pessimistic and optimistic robustness assessment with categorical inputs. Despite of the NP-hard nature of the problem, we show that a provably accurate assessment is approachable for any classifier using simple yet efficient greedy search. Our study is useful for assessing the model robustness before deploying it to real-world safety-critical applications. Our future work will pursue to provide tighter quality bound for assessing adversarial robustness assessment on categorical data.

## ACKNOWLEDGEMENTS

The research reported in this publication was partially supported by funding from King Abdullah University of Science and Technology (KAUST).

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

## A  PROOF FOR THEOREM 1

**Lemma 1** *Supposing A and B are two independent random variables and U is given by* $P_{UAB} = P_{AB}P_{U|AB}$:

$$I(U;A) + I(U;B) \leq 2I(U;A,B) \tag{11}$$

Proof: Given the definition of mutual information $I(X;Y) = H(X) - H(X|Y)$, we need to prove:

$$2H(A,B|U) \leq H(A|U) + H(B|U) \tag{12}$$

With the chain rule of conditional entropy:

$$H(A,B|U) = H(A|U) + H(B|A,U) \tag{13}$$

It indicates that $H(A,B|U) \leq H(A|U)$, as $H(B|U) \geq 0$. Similarly, we can derive $H(A,B|U) \leq H(B|U)$ as well. Combing both inequality relations, Eq.12 holds, which in turn proves Eq.11.

Considering the markov chain in Figure.1, we assume a class label Y and the training set $S$ are statistically independent. $f$ depends on S as $f = \mathcal{A}(S)$, where $\mathcal{A}$ is the training paradigm (could be randomized). The generation of the adversarial example $X'$ depends on both the clean feature vector $X$ and the targeted classifier $f$. This is consistent with the fact that generating adversarial samples needs to understand the local geometry property of the decision boundary of the targeted classifier. Once we get the perturbed feature vector $X'$, the classification output $Y' = f(X')$.

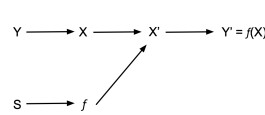

Figure 1: Markov chain of adversarial attack

We can derive the lower bound of the miss-classification probability with the perturbed categorical input $x'$ following Fano inequality:

$$\mathcal{R}_{\varepsilon}^{adv} = P(y' \neq y| \ |diff(b,\hat{b})| \leq \varepsilon) \ \geq \ 1 - \frac{I(Y';Y,S) + const}{\log K} \tag{14}$$

where $K$ denotes the number of the classes in the learning task. Based on Data Processing Inequality illustrated in Chapter 2 (Mack, 2006), we can further obtain:

$$\begin{aligned} \mathcal{R}_{\varepsilon}^{adv} \ &\geq \ 1 - \frac{I(X';Y,S) + const}{\log K}(I(Y';Y,S) \leq I(X';Y,S)) \\ &= 1 - \frac{2I(X;Y,S) - 2\eta_X - I(X';Y,S) + const}{\log K} \end{aligned} \tag{15}$$

where $\eta_X = I(X;Y,S) - I(X';Y,S)$ denotes the mutual information difference between the unperturbed $X$ and the adversarially perturbed instance $X'$. By applying Lemma.1 on $I(X';Y,S)$ in Eq.15, we have

$$\begin{aligned} \mathcal{R}_{\epsilon}^{adv} &\geq 1 - \frac{2I(X;Y,S) - 2\eta_X - I(X';S)/2}{\log K} \\ &\geq 1 - \frac{2I(X;Y,S) - 2\eta_X - I(f;S)/2}{\log K}(Data Processing Inequality, I(f;S) \geq I(X';S)) \end{aligned} \tag{16}$$

## B  EXPLANATION TO OBSERVATION.1

**First**, we follow the IID assumption in the definition of the expected adversarial risk in Eq.1. We assume $\ell$ adopts the form of zero-one loss function. If $f(\hat{x}) \neq y$, $\ell(f(\hat{x}) \neq y)) = 1$. Otherwise, $\ell = 0$. With this setting of the loss function, the expected adversarial risk of the classifier $f$ can be formulated as in Eq.17. This gives Eq.1 (the adversarial risk of $f$) in the main paper.

$$\mathcal{R}_{\varepsilon}^{adv} = \sup_{x \sim \mu_x, |\text{diff}(x,\hat{x})| \leq \varepsilon} P[f(x) \neq f(\hat{x})] \tag{17}$$

where $P$ denotes the probability measurement. In this sense, the expected adversarial risk of $\mathcal{R}_{\varepsilon}^{adv}$ can be explained intuitively as the maximum miss-classification probability over an adversarial sample $\hat{x}$, with the underlying data distribution $\mu_{x,y}$ and the attack budget limited by $\varepsilon$.

**Second**, Theorem.1 provides the information-theoretical lower bound of $\mathcal{R}_{\varepsilon}^{adv}$ with respect to the data distribution $\mu_{x,y}$. Indeed, for any given training data set $S$ sampled from the joint distribution $\mu_{x,y}$,

$S$ can be considered as a random variable. Similarly, given a deterministic training paradigm $\mathcal{A}$, the classifier $f = \mathcal{A}(S)$ (trained with $S$ using the training algorithm $\mathcal{A}$) can be also treated as a random variable in the hypothesis space. For a testing input $(x, y)$ sampled from the same distribution $\mu_{x,y}$ as $S$, deriving Theorem.1 follows Fano's inequality in information theory (Chapter 2 in (Mack, 2006)):

- For any $x \sim \mu_x$ and $(x, y) \sim \mu_{x,y}$, following Fano's inequality, we can produce the lower bound of miss-classification probability over the adversarial sample $\hat{x}$ generated from $x$ (based on the information of $y$ and $f$):

$$P(f(x) \neq f(\hat{x}) \mid x \sim \mu_x, |\text{diff}(x, \hat{x})| \leq \varepsilon) \geq 1 - \frac{2I(x;y,S) - 2\eta_x - I(f;S)/2 + \text{const}}{\log K} \quad (18)$$

where $f(x) = y$. The lower bound of the miss-classification probability is composed by three mutual information measurements among the random variables $(x, y)$, $S$ and $f$.

- By definition, the following inequality holds:

$$\mathcal{R}_\varepsilon^{adv} = \sup_{x \sim \mu_x, |\text{diff}(x,\hat{x})| \leq \varepsilon} P[f(x) \neq f(\hat{x})] \geq P(f(x) \neq f(\hat{x}) \mid x \sim \mu_x, |\text{diff}(x,\hat{x})| \leq \varepsilon)$$
$$\geq 1 - \frac{2I(x;y,S) - 2\eta_x - I(f;S)/2 + \text{const}}{\log K} \quad (19)$$

where $(x, y) \sim \mu_{x,y}$, $S \sim \mu_{x,y}$ and $f \in \mathcal{H}$.

From this perspective, **Theorem.1 establishes a lower bound of the expected adversarial risk of $f$**: Given a deterministic training paradigm $\mathcal{A}$, for any training data set $S$ and any testing input instance $(x, y)$ sampled from the underlying data distribution $\mu_{x,y}$, the miss-classification probability of $f$ over the adversarial sample generated from $x$ (based on $y$ and $f$) can be bounded by the three mutual information measurements $I(x;y,S)$, $\eta_x$ and $I(f;S)$, as in Eq.2 and 3 of Theorem.1 in the main paper.

**Comparison with the adversarial risk bound proposed by (Tu et al., 2019).** A relevant work of adversarial risk analysis can be found in (Tu et al., 2019), where the expected adversarial risk of a classifier $f$ is measured by the maximum classification loss over the distribution of adversarial samples (Definition.1 in (Tu et al., 2019)):

$$\mathcal{R}^{adv} = \mathbb{E}_{x,y \sim \mu_{x,y}} [ \sup_{\hat{x} \in N(x)} \ell(f(x'), y)] \quad (20)$$

where $N(x)$ denotes the neighbor set of $x$ derived by modifying features in $x$. The coverage of the neighboring set $N(x)$ can be measured by $l_p$ distance for continuous data or $l_0$ norm for discrete data. After that, Tu et al's work achieves two results. First, the expected adversarial risk bound can be bounded from above by the local worst-case risk of $f$ (noted as $\mathcal{R}_{\varepsilon, \mu_{x,y}}$):

$$\mathcal{R}^{adv} \leq \mathcal{R}_{\epsilon, \mu_{x,y}} = \mathbb{E}_{\hat{x} \sim \mu_{\hat{x}}, \mu_{\hat{x}} \sim \mathcal{B}_{W_p(\mu_x, \varepsilon)}} \ell(f(\hat{x}), y) \quad (21)$$

where $\mu_{\hat{x}}$ is a distribution sampled from the $p$-Wassernstein ball ($p \geq 1$) centered at $\mu_x$ and with a radius of $\varepsilon$. $N(x) \subseteq \mu_{\hat{x}}$. Second, the expected adversarial risk $\mathcal{R}_\varepsilon^{adv}$ can be bounded by the upper bound of the local worst-case risk of $f$ consisting of the rademarcher complexity of $f$ and the empirical risk of $f$, according to Theorem.1 and Remark.3 in (Tu et al., 2019). This upper bound holds with probability at least $1 - \delta$:

$$\mathcal{R}^{adv} \leq \hat{\mathcal{R}}_{P_n}^{adv} + 2\mathfrak{R}_n(\psi(f)) + M\sqrt{\frac{\log(1/\delta)}{2n}}$$
$$\hat{\mathcal{R}}_{P_n}^{adv} = \frac{1}{n} \sum_{i=1}^n \max_{\hat{x}_i \sim \mu_{\hat{x}}} \ell(f(\hat{x}_i, y_i)) \quad (22)$$

where $\hat{\mathcal{R}}_{P_n}^{adv}$ denotes the adversarial empirical risk calculated using $n$ IID samples from the adversarial data distribution $\mu_{\hat{x}}$. Practically, $\hat{\mathcal{R}}_{P_n}^{adv}$ can be estimated by calculating the maximum classification loss reached by perturbing $x$ within the attack budget. $M$ denotes the maximum output value of $f$. $\mathfrak{R}_n(\psi(f))$ denotes the expected rademarche complexity of an augmented function defined on $f$. The value of $\mathfrak{R}_n(\psi(f))$ is independent from the empirical adversarial risk $\hat{\mathcal{R}}_{P_n}^{adv}$. The formulation of $\mathfrak{R}_n(\psi(f))$ can be found on Page 10 of (Tu et al., 2019).

Our work differs from the results in (Tu et al., 2019) from the following perspective. The core contribution of (Tu et al., 2019) is to associate the Rademacher complexity of the classifier $f$ with $f$'s expected adversarial risk. Beyond that, for a classifier adopted in a learning scenario, we believe both the characteristics of input data and the classifier's complexity measurement determine its adversarial risk. As shown by the mutual information-based lower bound (Eq.2 and 3 of Theorem.1), the mutual information $I(f;S)$ measures the dependence of a trained $f$ over the training data $S$ sampled from $\mu_{x,y}$. $I(f;S)$ can be considered as the lower bound of the VC-dimension of $f$ (Xu & Raginsky, 2017; Zhu et al., 2020), which encodes the classifier's complexity. The other two mutual information measurements $I(x;y,S)$ and $\eta_x$ relates to the characteristics of the input data $(x,y)$. Intuitively, $I(x;y,S)$ measures the informativeness of $x$ for classification. A lower $I(x;y,S)$ denotes weaker consistency between the input data instance $(x,y)$ and the training data set $S$. $\eta_x$ reflects the sensitivity of the modified categorical features during the evasion attack. A classifier with input data containing more sensitive features is more likely to be compromised by modifying intentionally the values of the sensitive feature dimensions.

**Relations to the proposed robustness measurement.** Since it is not feasible to directly compute the expected risk $\mathcal{R}_\varepsilon^{adv}$, we empirically evaluate the adversarial risk/robustness and presented the results in Table 2 and Table 3 in Section 4.2 and 4.3. The empirical evaluation of the adversarial risk evaluation of $f$ over the data distribution $\mu_{x,y}$ is computed as a sample average,

$$\hat{\mathcal{R}}^{adv}(f) = \frac{1}{n}\sum_{i=1}^{n}\max_{\hat{x}_i,|\text{diff}(\hat{x}_i,x_i)|\leq\varepsilon}\ell(f(\hat{x}_i),y_i). \tag{23}$$

To facilitate computing, we adopt a smooth loss $\ell$ as a surrogate function approximating the zero-one loss. In Eq.(23), $\max_{\hat{x}_i,|\text{diff}(\hat{x}_i,x_i)|\leq\varepsilon}\ell(f(\hat{x}_i),y_i)$ is to estimate the maximum classification loss of $f$ over the adversarial sample $\hat{x}_i$, which is generated by modifying at most $\varepsilon$ categorical features of $x_i$. By its nature of set function maximization, $\hat{\mathcal{R}}^{adv}(f)$ is calculated in two difference settings: the pessimistic and optimistic robustness assessment defined in Eq.4 and Eq.5 in the main paper. When setting the threshold $\Gamma=0$ in Eq.4 in the main paper, the pessimistic estimate $\psi^{pess}$ (defined by $f_{y_k}-f_{y_K}$) is proportional to the classification loss, and gives the miss-classification confidence of the target classifier at $x$ with the attack budget limited by $\varepsilon$. this "miss-classification confidence" is the empirical estimator of the adversarial risk $f$ has a given input $x$, or called *robustness*.

## C    PROOF FOR THEOREMS ON THE WEAK SUBMODULARITY

We introduce the notion of $\gamma$-weak submodularity used in Theorem 2.

**Definition 3** *Monotone and $\gamma$-weakly submodular function (Santiago & Yoshida, 2020). A set function $g(S)$ is monotone, if for every $T \subset S$, we have $g(S) \geq g(T)$. Given a set cardinality threshold $k \geq 1$ and a scalar value $\gamma \in [0,1]$. We define $\gamma_k$ of $g(.)$ w.r.t. a set $S$ as:,*

$$\gamma_k = \min_{S\subseteq H, A:|A|\leq k, S\cap A=\emptyset}\frac{\sum_{a\in A}g(S\cup a)-g(S)}{g(S\cup A)-g(S)} \tag{24}$$

*If $g(S)$ is monotone and $\gamma_k > \gamma$ for any $k$, $g(S)$ is **a $\gamma$-submodular function**. Otherwise, $g(S)$ is $\gamma$-submodular, if $g(S)$ is non-monotone but $g(S)$ follows for any $k$:*

$$\sum_{a\in A}g(S\cup a)-g(S)\geq\{\gamma(g(S\cup A)-g(S)),1/\gamma(g(S\cup A)-g(S))\} \tag{25}$$

*It is easy to find, if $g(S)$ is monotone and $\gamma \geq 1$, $g(S)$ is submodular.*

Proof of deriving the robustness assessment score under the pessimistic scenario in Eq.7: we first formulate pessimistic scenario as a set function optimization,

$$S^* = \max_S\{\max_{k=1,2,3,...,K-1}\{F_k(S)\}+(G_K(S))\}$$
$$F_k(S) = \max_{l\subset S}f_k(b_l) \tag{26}$$
$$G_K(S) = \max_{l\subset S}\tilde{f}_K(b_l)$$

where $b_l$ denotes the modification of categorical features indicated by the index set $l$. $\tilde{f}_K(b_l) = -f_K(b_l)$. For each $k \in 1,2,3,...,K$, $\epsilon_k = f_k(q) - f_k(p) - \langle \nabla f_k(p), q - p \rangle$. $F_k(S)$ and $G_K(S)$ are monotonically non-decreasing set functions.

Supposing $\epsilon_k \geq 0$ for each $k \in \{1,2,3,...,K-1\}$, we assume further that the features indicated by $l'$ are modified in addition to $l$, with $|l'| \leq \zeta$. For any $b_l$, $b_{l \cup l'}$ and $b_{l \cup j}$ denote the additional modification over the features indicated by $l'$ and by $j$ respectively to increase the output of $f_k$ (increasing the miss-classification decision confidence) and $-f_K$ (decreasing the confidence of correct classification). In the following analysis, we relax the discrete indicator $b$ to the continuous domain, as each $b_i \in [0,1]$. $\nabla f_k(b_l)$ denotes the gradient of the classifier function $f_k$ with respect to the variable $b$ at $b = b_l$. Modifying from $b_l$ to $b_{l \cup l'}$ or $b_{l \cup j}$ follows the direction of gradient ascent, which gives $\langle \nabla f_k(b_l), b_{l \cup l' - b_l} \rangle \geq 0$ and $\langle \nabla f_k(b_l), b_{l \cup j - b_j} \rangle \geq 0$, $\langle \nabla \tilde{f}_K(b_l), b_{l \cup l' - b_l} \rangle \geq 0$ and $\langle \nabla \tilde{f}_K(b_l), b_{l \cup j - b_j} \rangle \geq 0$.

$$
\begin{aligned}
f_k(b_{l \cup l'}) - f_k(b_l) &\leq \langle \nabla f_k(b_l), b_{l \cup l'} - b_l \rangle + \frac{M_{k,\Omega_\zeta}}{2} \|b_{l \cup l'} - b_l\|_2^2 \\
&\leq \|\nabla f_k(b_l)_\zeta\|_2 + \frac{M_{k,\Omega_\zeta}}{2}|\zeta| \\
\sum_{j \in \zeta} f_k(b_{l \cup j}) - f_k(b_l) &\geq \sum_{j \in \zeta} \langle \nabla f_k(b_l), b_{l \cup j} - b_l \rangle + \frac{m_{k,\Omega_1}}{2} \|b_{l \cup j} - b_l\|_2^2 \\
&\geq \|\nabla f_k(b_l)_\zeta\|_2 + \frac{m_{k,\Omega_1}}{2}|\zeta| \; (f_k \text{is non-decreasing})
\end{aligned}
\tag{27}
$$

We can derive the lower bound of the submodularity ratio $\gamma_{k,\zeta}$ of $f_k$:

$$
\gamma_{k,\Omega_\zeta} = \frac{\sum_{j \in \zeta} f_k(b_{l \cup j}) - f_k(b_l)}{f_k(b_{l \cup l'}) - f_k(b_l)} \geq \frac{\|\nabla f_k(b_l)_\zeta\|_2 + \frac{m_{k,\Omega_1}}{2}|\zeta|}{\|\nabla f_k(b_l)_\zeta\|_2 + \frac{M_{k,\Omega_\zeta}}{2}|\zeta|}
\tag{28}
$$

Supposing $\epsilon_k < 0$ for each $k \in \{1,2,3,...,K-1\}$, we can derive:

$$
\begin{aligned}
f_k(b_{l \cup l'}) - f_k(b_l) &\leq \langle \nabla f_k(b_l), b_{l \cup l'} - b_l \rangle - \frac{m_{k,\Omega_\zeta}}{2} \|b_{l \cup l'} - b_l\|_2^2 \leq \frac{1}{2m_{k,\Omega_\zeta}} \|\nabla f_k(b_l)_\zeta\|_2^2 \\
\sum_{j \in \zeta} f_k(b_{l \cup j}) - f_k(b_l) &\geq \sum_{j \in \zeta} \{\langle \nabla f_k(b_l), b_{l \cup j} - b_l \rangle - \frac{M_{k,\Omega_1}}{2} \|b_{l \cup j} - b_l\|_2^2 \} \\
&\geq \|\nabla f_k(b_l)_{\Omega_\zeta}\|_2 - \frac{M_{k,\Omega_1}}{2}|\zeta| \; (f_k \text{is non-decreasing})
\end{aligned}
\tag{29}
$$

Given the smoothness assumption on the targeted classifier (See Definition.2), there exits a value of $M_{k,\Omega_1}|\zeta|/2 \leq \|\nabla f_k(b_l)_\zeta\|_2$, which allows that $\|\nabla f_k(b_l)_\zeta\|_2 - \frac{M_{k,\Omega_1}}{2}|\zeta| \geq 0$ holds. Therefore, we can derive the lower bound of the submodularity ratio of $F_k$ and $-G_K$, $\gamma_{k,\zeta}$ (k=1,2,3,...,K):

$$
\gamma_{k,\Omega_\zeta} = \frac{\sum_{j \in \zeta} f_k(b_{l \cup j}) - f_k(b_l)}{f_k(b_{l \cup l'}) - f_k(b_l)} \geq \frac{2m_{k,\Omega_\zeta}}{\|\nabla f_k(b)_v\|_2^2} (\|\nabla f_k(b)_v\|_2 - M_{k,\Omega_1}|\zeta|/2)
\tag{30}
$$

The submodularity ratio $\gamma_\zeta$ on $\Omega_\zeta$ in Eq.26 is $\gamma_\zeta = \min_{k=1,2,3,...,K} \{\gamma_{k,\Omega_\zeta}\}$.

Proof of deriving the robustness assessment score under the optimistic scenario in Eq.8. Following the setting of Eq.26, we can formulate the robustness assessment problem at the optimistic scenario (with an oblivious adversary) :

$$
\begin{aligned}
S^* &= \max_S \{G_K(S) + \min_{k=1,2,3,...,K-1} \{\tilde{F}_k(S)\}\} \\
\tilde{F}_k(S) &= \min_{l \subset S} h_k(b_l) \\
G_K(S) &= \min_{l \subset S} f_K(b_l)
\end{aligned}
\tag{31}
$$

Both $G_k(S)$ and $\tilde{F}_k(S)$ are non-increasing with $h_k(b_l) = -f_k(b_l)$. Eq.31 thus defines a non-monotone set function maximization problem (See Definition.3). We conduct the analysis as in Eq.27 and Eq.29. Modifying $b_l$ to $b_{l \cup l'}$ and $b_{l \cup j}$ follows the direction of gradient descent, which gives $\langle \nabla f_K(b_l), b_{l \cup j} - b_l \rangle \leq 0$ and $\langle \nabla f_K(b_l), b_{l \cup l'} - b_l \rangle \leq 0$, $\langle \nabla h_k(b_l), b_{l \cup j} - b_l \rangle \leq 0$ and $\langle \nabla h_k(b_l), b_{l \cup l'} - b_l \rangle \leq 0$.

Supposing $\epsilon \geq 0$, we can derive:

$$f_K(b_{l \cup j}) - f_K(b_l) \geq \langle \nabla f_K(b_l), b_{l \cup j} - b_l \rangle + \frac{m_{K,\Omega_1}}{2} \|b_{l \cup j} - b_l\|_2^2$$

$$f_K(b_{l \cup j}) - f_K(b_l) \leq \langle \nabla f_K(b_l), b_{l \cup j} - b_l \rangle + \frac{M_{K,\Omega_1}}{2} \|b_{l \cup j} - b_l\|_2^2$$

$$f_K(b_{l \cup l'}) - f_K(b_l) \geq \langle \nabla f_K(b_l), b_{l \cup l'} - b_l \rangle + \frac{m_{K,\Omega_\zeta}}{2} \|b_{l \cup l'} - b_l\|_2^2$$

$$f_K(b_{l \cup l'}) - f_K(b_l) \leq \langle \nabla f_K(b_l), b_{l \cup l'} - b_l \rangle + \frac{M_{K,\Omega_\zeta}}{2} \|b_{l \cup l'} - b_l\|_2^2$$

(32)

and

$$h_k(b_{l \cup j}) - h_k(b_l) \leq \langle \nabla h_k(b_l), b_{l \cup j} - b_l \rangle - \frac{m_{k,\Omega_1}}{2} \|b_{l \cup j} - b_l\|_2^2$$

$$h_k(b_{l \cup j}) - h_k(b_l) \geq \langle \nabla h_k(b_l), b_{l \cup j} - b_l \rangle - \frac{M_{k,\Omega_1}}{2} \|b_{l \cup j} - b_l\|_2^2$$

$$h_k(b_{l \cup l'}) - h_k(b_l) \leq \langle \nabla h_k(b_l), b_{l \cup l'} - b_l \rangle - \frac{m_{k,\Omega_\zeta}}{2} \|b_{l \cup l'} - b_l\|_2^2$$

$$h_k(b_{l \cup l'}) - h_k(b_l) \geq \langle \nabla h_k(b_l), b_{l \cup l'} - b_l \rangle - \frac{M_{k,\Omega_\zeta}}{2} \|b_{l \cup l'} - b_l\|_2^2$$

(33)

From Eq.32, we conduct the similar analysis as in Eq.22 and 23, which gives the submodularity ratio at the optimistic case:

$$\sum_{j \in l'} f_K(b_{l \cup j}) - f_K(b_l) \geq \frac{m_{K,\Omega_1}|\zeta|}{2} - \|\nabla f_K(b_l)_\zeta\|_2$$

$$f_K(b_{l \cup l'}) - f_K(b_l) \leq \frac{M_{K,\Omega_\zeta}|\zeta|}{2} - \|\nabla f_K(b_l)_\zeta\|_2$$

(34)

$$\sum_{j \in l'} h_k(b_{l \cup j}) - h_k(b_l) \geq \sum_{j \in l'} \langle \nabla h_k(b_l), b_{l \cup j} - b_l \rangle - \frac{M_{k,\Omega_1}}{2} \|b_{l \cup j} - b_l\|_2^2 \geq -\frac{M_{k,\Omega_1}|\zeta|}{2} - \|\nabla h_k(b_l)_\zeta\|_2$$

$$h_k(b_{l \cup l'}) - h_k(b_l) \leq \langle \nabla h_k(b_l), b_{l \cup l'} - b_l \rangle - \frac{m_{k,\Omega_\zeta}}{2} \|b_{l \cup l'} - b_l\|_2^2 \leq \frac{-m_{k,\Omega_\zeta}|\zeta|}{2} - \|\nabla h_k(b_l)_\zeta\|_2$$

(35)

Let $\gamma_k \in (0,1)$ and $-\frac{M_{k,\Omega_1}|\zeta|}{2} - \|\nabla h_k(b_l)_\zeta\|_2 \geq \frac{1}{\gamma_k}(\frac{-m_{k,\Omega_\zeta}|\zeta|}{2} - \|\nabla h_k(b_l)_\zeta\|_2)$. The submodularity ratio of $h_k$, $\gamma_k \leq \frac{\frac{m_{k,\Omega_\zeta}|\zeta|}{2} + \|\nabla h_k(b_l)_\zeta\|_2}{\frac{M_{k,\Omega_1}|\zeta|}{2} + \|\nabla h_k(b_l)_\zeta\|_2}$.

Proofs to Theorem.3 (Eq.9). In the pessimistic scenario, assessing adversarial robustness can be formulated as a problem of monotone weakly submodular function maximization. Assuming that $m_f(l)$ is the solution $l$ obtained by the *FSGS* method and $m_f(l^*)$ is the optimal solution $l^*$ to Eq.4. Based on Theorem.3 in (Elenberg et al., 2018), we can obtain:

$$1 - |m_f(l)| \geq (1 - |m_f(l^*)|)(1 - e^{-\gamma_\zeta^{pess}})$$

(36)

Therefore, we can derive the first inequality (the pessimistic scenario) in Eq.9

$$|m_f(l)| + e^{-\gamma_\zeta^{pess}} \leq (1 - e^{-\gamma_\zeta^{pess}})|m_f(l^*)|$$

(37)

Assessing adversarial robustness in the optimistic scenario (Eq.5) is in nature a problem of non-monotone weakly submodular function maximization. Assuming that $m_f(l)$ is the greedy search based solution obtained by the *RandGS* (Randomized Greedy Search) and $m_f(l^*)$ is the optimal solution to Eq.5. Applying Theorem 1.9 (Santiago & Yoshida, 2020) directly, we can obtain:

$$|m_f(l)| \geq \frac{1}{\gamma_\zeta^{optim}} e^{-\gamma_\zeta^{optim}} |m_f(l^*)|$$

(38)

The proof to the approximation gap of *OMPGS* in Eq.10 follows the proof to Theorem 2.4 and Theorem 3 (Santiago & Yoshida, 2020).

# D  ALGORITHM PSEUDO CODES FOR FSGS AND RANDGS BASED ROBUSTNESS ASSESSMENT

---

**Algorithm 1** FSGS for the pessimistic robustness assessment

---

**Input:** The candidate set $H=\{a_i\}_{i=1\ldots n}$ of all modifiable features
**Output:** The maximum support set $l_i$ pushing $m_f(l_i)$ to violate the tolerance constraint in Eq. 4
  1: $l_0 \leftarrow \emptyset$
  2: **for** $t=0,1,2,\ldots,\rho$ **do**
  3:     $T \leftarrow$ *zero-valued vector* $\in \mathbb{R}^n$
  4:     **for** each $a_i \in H/l_{t-1}$ **do**
  5:         $T(a_i) = \underset{s \subset l_{t-1}}{\arg\max}\ m_f(s \cup a_i)$ (***Inner level Optimization***)
  6:     **end for**
  7:     $\tilde{a} = \underset{a_i}{\arg\max}\ T(a_i)$ (***Outer level Optimization***)
  8:     $l_i \leftarrow l_{i-1} \cup \{\tilde{a}\}$
  9:     $m_f(l_i) \leftarrow m_f(l_{i-1} \cup \tilde{a})$
 10:     **if** $m_f(l_i) \geq \Gamma$ **then break**
 11: **end for**

---

**Algorithm 2** RandGS for the optimistic robustness assessment

---

**Input:** The candidate set $H=\{a_i\}_{i=1\ldots n}$ of all modifiable features
**Output:** The maximum support set $l_i$ pushing $m_f(l_i)$ to violate the tolerance constraint in Eq. 5
  1: $l_0 \leftarrow \emptyset$
  2: **for** $t=0,1,2,\ldots,\rho$ **do**
  3:     $T \leftarrow$ *zero-valued vector* $\in \mathbb{R}^n$
  4:     **for** each $a_i \in H/l_{t-1}$ **do**
  5:         $T(a_i) = \underset{s \subset l_{t-1}}{\arg\max}\ m_f(s \cup a_i)$ (***Inner level Optimization***)
  6:     **end for**
  7:     $M \subset H/l_{t-1}$ *be a subset of size* $\tau$
  8:     *minimizing* $\sum_{a_i \in M} T(a_i)$ (***Outer level Optimization***)
  9:     *Let* $\tilde{a}$ *be a uniformly random element from* $M$
 10:     *Let* $\tilde{s}(\tilde{a})$ *be the subset corresponding to* $\tilde{a}$
 11:     $l_i \leftarrow l_{i-1} \cup \{\tilde{a}\}$
 12:     $m_f(l_i) \leftarrow \min\{m_f(\tilde{a} \cup \tilde{s}(\tilde{a})), m_f(l_{i-1})\}$
 13:     **if** $m_f(l_i) \geq \Gamma$ **then break**
 14: **end for**

---

# E  ORTHOGONAL MATCHING PURSUIT GREEDY SEARCH

The pseudo-codes of *OMPGS* is presented in Algorithm 3, which explains how it is adopted to solve Eq.4 and Eq.5. In each iteration, for each subset $s$ of the previous support set $l_{k-1}$, we compute the gradient of $f_y$ with respect to $\mathbf{b}_s$ (binary indicator matrix $\mathbf{b}$ with the entries in $s$ changed). The top-$k'$ attributes with the largest magnitudes in the gradient vector are selected to form a candidate set $\hat{s}$. We can then find out the optimal attribute $j \in \hat{s}$ to extend for each subset $s$ and record the optimal attack objective value $g(l_{k-1} \cup j)$. In the outer iteration, we choose finally the attribute $j^*$ producing the largest marginal gain to add into $l_{k-1}$.

The worst case cost of objective function evaluation in each iteration of *OMPGS* is bounded by $O((\sum_{i=1}^{k-1} C_k^i))|k'|$. We adjust $k'$ to achieve a trade-off between enlarging the search range of greedy search and the cost of function evaluation. Usually $k'$ is much less than $|H|-|l_k|$, especially when $H$ is significantly large (e.g. $H > 10k$). Thus *OMPGS* runs significantly faster than *FSGS*.

We further illustrate the rationality of using the orthogonal matching pursuit step in *OMPGS* to solve the weakly submodular maximization based assessment problem. In Eq. 39, we unveil that the gradient

---

**Algorithm 3** Orthogonal Matching Pursuit based Greedy Search (OMPGS)

---

**Input:** The attack budget $K$, the set function based attack objective $g(l)$ and the set $H = \{(i,j), i = 1...n, j = 1...m\}$ of all the modifiable discrete attributes selected support set $l_k$, with $|l_k| \leq K$;
**Output:** $g(l_k)$ and the optimal subset of $l_k$ achieving the attack goal $S_0 \leftarrow \emptyset$

1: **for** $k = 1,2,...,K$ **do**
2:     $T = \emptyset$
3:     **for** $s \subset l_{k-1}$ **do**
4:         $r \leftarrow \nabla f_y(\mathbf{b}_s)$
5:         $\hat{s} = \{j_1, j_2,...,j_{k'}\} \leftarrow \underset{j \in \{H/l_{k-1}\}}{\operatorname{argmax}} |<e_j, r>|$
6:         $\tilde{j} \leftarrow \underset{j \in \hat{s}}{\operatorname{argmax}} g(l_{k-1} \cup \{j\})$
7:         $T = T \cup \{(\tilde{j}, g(l_{k-1} \cup \tilde{j}))\}$
8:     **end for**
9:     $j^* \leftarrow \underset{j \in T}{\operatorname{argmax}} g(l_{k-1} \cup \{j\})$
10:    $l_k \leftarrow l_{k-1} \cup \{j^*\}$
11: **end for**

---

magnitude of $f_y$ with respect to the binary indicator variables $b_i^j$ can be used as an estimator of the marginal gain of $m_f$ in each iteration of the greedy search of *OMPGS*.

We assume that $\hat{b}$ and $\hat{b}'$ denote two sets of categorical attribute changes. $b$ indicates the category value assignment of an unperturbed data instance $\hat{x}$. $|\text{diff}(b,\hat{b})| \leq \zeta, |\text{diff}(b,\hat{b}')| \leq \zeta, |\text{diff}(\hat{b},\hat{b}')| \leq \zeta, \zeta \geq 1$. The targeted classifier is denoted as $f_{y_k}(x)$ (k=1,...,$\mathbf{K}$), in which $\mathbf{K}$ is the correct class label.

$$|f_{y_k}(\hat{x},\hat{b}') - f_{y_k}(\hat{x},b)| \leq \max\{\frac{1}{2m_{k,\zeta}}\|\nabla f_{y_k}(x,\hat{b})_\nu\|_2^2, \|\nabla f_{y_k}(x,\hat{b})_\nu\|_2 + M_{k,\Omega_\zeta}|\zeta|/2\}$$

$$|f_{y_\mathbf{K}}(\hat{x},\hat{b}') - f_{y_\mathbf{K}}(\hat{x},\hat{b})| \geq \min\{\frac{1}{2M_{\mathbf{K},\zeta}}\|\nabla f_{y_\mathbf{K}}(x,\hat{b})_\nu\|_2^2, \|\nabla f_{y_\mathbf{K}}(x,\hat{b})_\nu\|_2 + m_{k,\Omega_\zeta}|\zeta|/2\}$$

(39)

where $\nu = \text{diff}(\hat{b},\hat{b}')$.

## F    EXPERIMENTAL EVALUATION

We first introduce the evaluation datesets in section F.1, the experimental setup in section F.2, and the complexity analysis and running time in section F.3. Besides the multi-class datasets *Yelp*, *IPS* and *Splice*, we also include another two binary classification datasets collected from *Cyber Security* and *HealthCare* in the experiments. We add the feature sensitivity with the multi-class datasets *Yelp*, *IPS* and *Splice* in F.4 as the supplementary information to the main text. We add the the robustness assessment with the binary-class datasets *PEDec* and *EHR* in F.5. It includes the summarised table on the robustness assessment and the information-theoretic characterization of adversarial vulnerability verified in the experiment.

### F.1    DATASET INFORMATION

**Yelp-5** (**Yelp**)(Asghar, 2016). We use the Yelp-5 provided in the torchtext package contains 650,000 training and 50,000 testing textual samples. In the classifier training process. We learn a 300-dimensional embedding for each word. Thus, each sentence is encoded as $x \in \mathbb{R}^{n*300}$, where $n$ is the number of words. we treat every word as a categorical feature and put this data into the LSTM model with five level rating classes of the reviews as the labels. The task is to predict the 5-class rating of each review. After learning the classifier, we start to attack the categorical input with modifying the words with those semantic neighbor words, which can contain the sentence is semantically correct. Here not all the words have neighbors. Here we choose the 200 data from the testing data for attack.

**Intrusion Prevention System Dataset (IPS)**(Wang et al., 2020). An adversary of network intrusion performs a series of actions to compromise the targeted devices. We collect one day of IPS records via a security service vendor from 242,467 endpoint devices. After removing repeated instances, the

collected data set contains 4,101 time series of attack events. Each sequence instance is composed by 20 attack steps. On each attack step, the adversary can choose one of 1,103 different malicious actions registered as highly threatening ones. Thus one data instance of IPS data is given as $\mathbf{x} \in R^{20*1103*70}$ according to the definition given in Section.3. Each of the 1103 attack actions is projected to a 70-dimensional embedding vector. In our study, each sequence instance is used as input to the classification system, which predicts the most likely attack operation conducted at the immediately successive step of the sequence. We randomly select 80% of each dataset for training and others for testing. Based on the prediction output, security analysts can proactively take prevention actions. We focus on predicting the occurrence of the two most threatening actions related to recently uncovered vulnerability. We thus study a 3-class classification task: the 2 highly malicious actions and all the others as the third class. All the privacy-related information, such as names of the devices, IP addresses and hosting machines' domain names have been removed and anonymized.

**Splice-junction Gene Sequences (Splice)** (Noordewier et al., 1991). Splice junctions are the points on a DNA sequence at which redundant DNA segments are removed during the process of protein creation. As shown in Figure 2 (Htike & Win, 2013), a precursor mRNA contains exons and introns. During splicing, introns get spliced out while exons get reunited. It is then important to analyze the role of splice junction sequences in mRNA. The *Splice* dataset contains 3,190 samples from Genbank release 64.1. Each of the

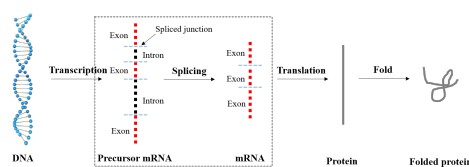

Figure 2: RNA synthesis and translation into protein.

instance is a gene fragment of 60 categorical features, which consists of a window that covers 30 nucleotides respectively before and after each donor and acceptor site. Each categorical feature can take one among 'A', 'G', 'C', 'T' or 'N'(stands for ambiguity). In our work, Each category is cast to a 30-dimension embedding. Then one data instance of the dataset is given as $\mathbf{x} \in R^{60*5*30}$. The dataset defines a 3-class classification task. Each gene instance can be an intro-exon boundary (labelled as *IE*), an exon-intro boundary (labelled as *EI*), or any other cases (labelled as *N*). There are 768 *IE* instances and 767 *EI* instances in total. All the rest are *N* instances. In the published dataset, the data records are anonymized. We randomly select 80% of each dataset for training and others for testing.

**Windows PE Malware Detection (PEDec).** We build a well-balanced dataset composed by full dynamic analysis reports of 20000 benign and malicious PE executables. This dataset is used for a binary classification scenario of PE malware detection. We collect 10,972 malware samples of 152 families randomly selected from those submitted to VirusTotal between 2018 and 2020. Each of the files https://www.overleaf.com/project/5f81ef055ce9280001ecdee8is classified as malicious by more than 21 antivirus engines. We further use AVClass to ensure that no malware families are over-represented in the collected data. For the benign samples, we choose the fresh installation packages of all the community-maintained packages of Chocolatey, where the package submissions go through a strict review process to exclude malicious and pirated software. Therefore we obtain 9028 benign samples of third-party and Windows system files. Each executable of the dataset is encoded into a binary feature vectors with 5000 signatures selected by human experts. Each signature covers presence or absence of specific windows API calls, URL access, registry table edits, file I/O operations and the status of IP ports. The ensemble of the signatures' binary states form a description of the file's behavior in execution within a sandbox. In our study, we use the 5000 binary codes directly as the features of a file instance. Simple as it is, the combination space of these signatures can still induce prohibitively expensive search cost, which prevent brute-force attacks. We randomly select 80% of each dataset for training and others for testing.

**Electronic Health Records (EHR)** (Ma et al., 2018). The real-world EHR dataset consists of time-ordered medical visit records of 7314 patients. Each patient has from 4 to 200 medical visits. Each visit record is composed by a subset of 4130 discrete ICD9 diagnosis codes[2]. Each diagnosis code represents occurrence of a disease, a symptom, or an abnormal finding. Using the historical EHR data of patients, we can predict the risk of patients suffering the target diseases. In this experiment, our target is a binary classification task: we forecast whether a patient will suffer heart failure disease in the future. In our experiments, a data instance of EHR data set is organized as a tensor $\mathbf{x} \in R^{200*4130*70}$ with

---

[2]http://www.icd9data.com/

each of the 4130 diagnosis codes projected to a 70-dimensional embedding vector. For the patients with less than 200 visits, we pad the empty observations by setting the corresponding $b_i^j = 0$.

We split randomly each dataset into two non-overlapped subsets: 80% of them are used for training and the left 20% form a testing set. In the testing set, we choose the correctly classified testing samples with the strongest probabilistic confidence provided by the classifier to perform evasion attack. There are thus 125, 500, 324, 3000 and 414 instances of *Yelp*, *IPS*, *Splice*,*PEDec* and *EHR* used in the evasion attack scenario. On *PEDec* and *EHR* data, since the discrete feature takes binary values, we focus on adversarial modification by simply flipping the codes. For *Yelp*, *IPS* and *Splice*, the adversarial attack is conducted by replacing the original category values of a targeted discrete attribute with a new one.

## F.2    EXPERIMENTAL SETUP

We instantiate the study of robustness characterization and assessment to the popularly deployed Convolution Neural Networks (CNN) and Long Short-Term Memory (LSTM) models. For *PEDec* dataset, we adopt a simple CNN model composed of one convolution layer followed by two linear layers. The rest datasets contain sequential instances, we thus apply standard LSTM as the classifier in the experiments. Without loss of generality, we use *ReLu* activation function in both the CNN and LSTM classifier with the dropout module. Theorem.2 applies in both cases. Both models define Lipschitz-continuous decision functions, which meets the smoothness constraint given in Definition 2. For the instances used for robustness assessment, the CNN and LSTM classifier achieve accurate classification over all the datasets. They achieve accuracy scores of *0.90*, *0.95*, *0.94*, *0.93* and *0.62* respectively for the datasets *IPS*, *Splice*, *PEDec*, *EHR* and *Yelp*.

In Section 4.1, we demonstrate robustness assessment of the two classifiers over all the datasets with varied tolerance levels against adversarial perturbation. We set the robustness threshold $\Gamma$ to $-0.4$ and $0$ respectively. The former defines a low tolerance to the decision perturbation caused by adversarial attack. A minor fluctuation of the classifier's output triggers an alert of potential evasion attack. In contrast, the latter setting only raises an alarm when the classifier produces an ambiguous decision close to the boundary or report a miss-classification incident. Furthermore, we show the derived robustness assessment scores with respect to an *oblivious adversary* (*OA*,the optimistic assessment) and an *knowledgeable adversary* (*KA*,the pessimistic assessment). The purpose is to show that 1) *our proposed robustness assessment method can adapt varied tolerance settings to adversarial threats in safety-critic applications*; 2) *we aim at confirming the impact of different query capabilities of the adversary over the robustness assessment scores*, as revealed by Theorem 2. For the instances used for robustness assessment, we report the median and mean of the derived robustness assessment scores. Both *FSGS/RandGS* and *OMPGS* are used to calculate the robustness assessment scores. They verify the effectiveness of greedy search for robustness assessment with categorical input. We implement the empirical study using the Python library PyTorch and conduct all the experiments on Linux server with 2 GPUs (GeForce 1080Ti) and 16-core CPU (Intel Xeon). For *RandGS* and *OMPGS*, we empirically set the number of candidate attributes in each iteration of greedy search to be 10 globally for all the datasets.

## F.3    TIME COMPLEXITY AND RUNNING TIME COMPARISON

We provide the time complexity analysis in Table.4. We assume a data instance $x$ containing $N$ categorical features and each feature can choose any of the $M$ categorical values. Furthermore, we assume *FSGS*, *OMPGS* and *GradAttack* all run $T$ iterations. Table.4 shows the query complexity of these methods of $T$ iterations.

In the table, $K_{\text{ompgs}}$ for *OMPGS* denotes the number of top-ranked candidate features in each iteration. *OMPGS* then selects one feature from the top $K_{\text{ompgs}}$ candidate features. The selected feature achieves the best attack performance by combing this feature and any subset of the already modified features. $K_{\text{grad}}$ for *GradAttack* denotes the number of the candidate features selected to modify in each iteration. As seen in Table.4, *FSGS* has the largest query complexity in each iteration. Enjoying the heuristics provided by the orthogonal matching pursuit step, *OMPGS* locates simultaneously the top-ranked candidate features and the categorical values of these candidate features that are likely to bring the most increase of $m_f$. Therefore, *OMPGS* can reduce the query complexity significantly compared to *FSGS*. *GradAttack* does not evaluate the gain of $m_f$ on the combination of the candidate features and

Table 4: Query complexity of *FSGS*, *OMPGS* and *GradAttack*

| Assessment Methods | Query complexity |
|---|---|
| FSGS | $\sum_{t=0}^{T}((N-t)*M*2^{t})$ |
| OMPGS | $\sum_{t=0}^{T}(K_{\mathrm{ompgs}}*2^{t})$ |
| GradAttack | $T*\sum_{k=0}^{K}(C_{K_{\mathrm{grad}}}^{k}*M^{k})$ |

Table 5: Averaged running time of *FSGS*, *OMPGS* and *GradAttack* on successfully attacked data instances (measured in seconds)

| Greedy Search | Yelp | IPS | Splice | EHR |
|---|---|---|---|---|
| FSGS | 1.62 | 263.8 | 13.7 | 446 |
| OMPGS | 0.82 | 0.43 | 0.08 | 27.44 |
| GradAttack | 0.18 | 10.60 | 1.07 | 4.05 |

Table 6: Average number of iterations of *FSGS*, *OMPGS* and *GradAttack* on successfully attacked data instances

| Greedy Search | Yelp | IPS | Splice | EHR |
|---|---|---|---|---|
| FSGS | 1.9 | 2.4 | 3.0 | 1.7 |
| OMPGS | 2.5 | 2.2 | 5.2 | 2.0 |
| GradAttack | 38.6 | 3.3 | 4.8 | 8.0 |

the subsets of the already modified features. The query complexity of *GradAttack* in each iteration only depends on $K_{\mathrm{grad}}$ and $M$.

We also show the running-time evaluation of these methods in Table.5 and Table.6. In both tables, we focus on the pessimistic robustness problem and set the threshold $\Gamma = 0$ in Eq.4 in the main paper. Table.5 records the averaged running time for computing robustness assessment results over the input data instances where the evasion attack is delivered successfully, i.e. $m_f$ increases above 0. Table.6 shows the average number of iterations required over the input instances to make $m_f$ increase over 0.

As seen in the two tables, *FSGS* stops the loops of greedy search generally earlier than *OMPGS* and *GradAttack*, while costing the largest running time. The observation is consistent with the design of *FSGS*. On one hand, *FSGS* traverses all the candidate features and chooses the one that most improves the marginal gain. Compared to *OMPGS* and *GradAttack*, *FSGS* has a better chance to achieve higher marginal gain of $m_f$ in each iteration. Therefore, *FSGS* requires fewer loops in the greedy search and modifies less number of features to solve the robustness assessment problem. However, on the other hand, *FSGS* makes larger query costs in each iteration of the greedy search. The running time of *FSGS* is thus higher than the other two methods.

On the contrary, *GradAttack* runs fast but needs significantly more iterations in the search loops. In each iteration, *GradAttack* only needs to evaluate the combinations of the $K_{grad}$ features selected from the candidate set and choose the combination contributing the best increase of $m_f$. It requires much fewer queries than *FSGS*, as it only searches a subset of the combinations that *FSGS* looks into. However, *GradAttack* likely results in less optimal solutions in each iteration. Therefore, *GradAttack* needs more loops to extend its search range.

*OMPGS* achieves the best trade-off in between. The orthogonal matching pursuit step adopted in *OMPGS* helps narrow down drastically the search range while guaranteeing the optimality of the features in the reduced search range. Compared to *FSGS*, *OMPGS* stops early the search with a similar number of loops, yet costing much fewer queries and less running time than *FSGS*. Compared to *GradAtttack*, *OMPGS* has a better chance to select the really useful feature modifications, thus costing fewer loops.

## F.4 FEATURE SENSITIVITY

To further understand the association between the feature sensitivity and the derived robustness assesment scores, we conduct one-factor-at-a-time sensitivity analysis (Campbell et al., 2008) to measure the sensitivity of features. Given a data instance, we change each feature while keeping all the others fixed. The averaged change of the classifier's output over all the testing instances is used as the feature-wise sensitivity measurement. A larger value indicates that the classifier's output is more

Table 7: The feature sensitivity of different datasets

| DataSet | Top3 | | | Top5 | | |
|---|---|---|---|---|---|---|
| | max | average | min | max | average | min |
| **IPS** | 1.00 | 0.53 | 0.00 | 1.00 | 0.53 | 0.00 |
| **Splice** | 1.00 | 0.38 | 0.00 | 1.00 | 0.33 | 0.00 |

Table 8: Top sensitive features vs frequently perturbed features

| DataSet | Yelp | IPS | Splice |
|---|---|---|---|
| **Sensitive** | **not**, but, we, could,**They** | **19,0,17**,18,16 | **29,28,30,31**,27 |
| **Perturbed** | day,never, **They**,good,**not** | **19,0**,1,2,**17** | **29,28,30,31**,32 |

sensitive to the change over the corresponding feature. In Table 7, we show the *maximum*, *average* and *minimum* decrease of the probabilistic output for *IPS* and *Splice*, induced by changing the **top-3** and **5**-ranked features.

First of all, we can find that the feature sensitivity of *IPS* lies in a narrow interval lower than all the other datasets. This resonates with the larger robust assessment scores in Table 1 and 2. Second, *Splice* also has a narrow interval with high sensitivity values, indicating a potential high vulnerability. Third, within dataset *IPS* and *Splice*, the sensitivity levels vary strongly, between the max and min value. That implies, changing the top sensitive features only is enough to deliver a successful attack. In real-world applications, the existence of such highly sensitive yet sparse sensitive attributes is widely witnessed. For example in *IPS*, the **top-5** ranked features are relevant to the key signatures of network traffic flows, and they are thus usually used by human security analysts to filer out suspicious incidents. The rest of the features in *IPS* are not explicitly relevant with the network I/O behaviours, which are less informative in the classification task. In *Splice*, the **top-5** sensitive features are found as the gene segments around the spliced junctions. They are particularly sensitive to the gene types. The rest of the features show little physical significance in the gene splicing process.

We also find that the top-sensitive features also appear as the most frequently selected features by *FSGS*,*GradAttack* and *OMPGS*. We further count the frequency of appearance for each feature selected in the pessimistic robustness assessment and compare them with the top-sensitive features, as given by Table 8. For *IPS*, the **top-3** sensitive features also appear in the list of the most frequently selected features. The **top-4** sensitive features in *Splice* are also the most frequently selected 4 features. The interesting overlapping between the attributes which are useful for attack and the top-ranked sensitive attributes confirms our intuition about the association between feature sensitivity and adversarial vulnerability of the classifier. On *Splice*, 4 of the most frequently attacked features (29,28,30 and 31 in Table 8) correspond to the junction of exons, which are also highly informative for gene categorization (Htike & Win, 2013). On *IPS*, 3 of the 5 most frequently attacked categorical features (19, 0 and 17) are the also informative indicators of unusual DNS requests and DDos activities. They are ranked as the most sensitive features according to Table 8.

Suppressing feature sensitivity helps improve adversarial robustness. However, these sensitive attributes are usually informative for classification in an adversary-free scenario. Excluding the sensitive features causes accuracy loss. It indicates the balance between classification utility and adversary-resilience feature engineering in ML practices.

## F.5 ROBUSTNESS ASSESSMENT IN BINARY CLASSIFICATION

### F.5.1 ROBUSTNESS ASSESSMENT WITH GREEDY SEARCH

Table 9 shows the results of robustness assessment for models built on binary-class datasets *PEDec* and *EHR*. We have the same findings in that with the multi-class datasets *Yelp*, *IPS* and *Splice*. In details, firstly for both the pessimistic and optimistic robustness assessment problems, a lower tolerance threshold $\Gamma$ results in lower scores over all the binary-class datasets. Secondly the pessimistic robustness assessment scores are always lower or equal to the optimistic assessment scores.

### F.5.2 ROBUSTNESS VS FEATURE INFORMATIVENESS

We take the $x_{l_{mi}}$ samples from the 62, 43 original samples for the dataset *PEDec*, *EHR* respectively on the half-way of pushing the $m_f$ of the original samples (*OS*) to surpass $\Gamma$. In Table 10, we verified

Table 9: Pessimistic and Optimistic Robustness Assessment Score $|l|$, reported with median and mean. A lower/higher value at a given percentile level indicates stronger/weaker robustness of the classifier. SR is the success rate of attack. OVF indicates SR=0.

| GreedySeearch | P/O | $\Gamma$ | PEDec | | EHR | |
|---|---|---|---|---|---|---|
| | | | SR | Med (Avg) | SR | Med (Avg) |
| FSGS | P | -0.4 | 0.92 | 3 (3.2) | 0.98 | 1 (1.2) |
| | | 0 | 0.87 | 3 (3.7) | 0.94 | 2 (1.2) |
| | O | -0.4 | 0.46 | 4 (4.1) | 0 | OVF |
| | | 0 | 0.42 | 4 (4.7) | 0 | OVF |
| GradAttack | P | -0.4 | 0.56 | 7 (7.2) | 0.63 | 2 (2.0) |
| | | 0 | 0.51 | 8 (6.1) | 0.62 | 2 (2.3) |
| OMPGS | P | -0.4 | 0.87 | 6 (6.1) | 0.95 | 2 (2.0) |
| | | 0 | 0.82 | 6 (6.3) | 0.94 | 2 (2.4) |
| | O | -0.4 | 0.62 | 5 (6.1) | 0.38 | 2 (2.7) |
| | | 0 | 0.54 | 6 (6.8) | 0.42 | 2 (2.5) |

Table 10: Robustness Assessment vs Feature Informativeness, reported at $\Gamma=0$. $I(x_{l_{mi}};y,S) < I(x;y,S)$ results in higher adversarial risk (higher SR, smaller $|l|$) on $x_{l_{mi}}$ than original $x$.

| GreedySeearch | P/O | sample | PEDec | | EHR | |
|---|---|---|---|---|---|---|
| | | | SR | Med(Avg) | SR | Med(Avg) |
| FSGS | P | original $x$ | 0.87 | 3 (3.7) | 0.94 | 2 (1.2) |
| | | $x_{l_{mi}}$ | 0.96 | 3 (3.1) | 1 | 1 (1) |
| | O | original $x$ | 0.42 | 4 (4.7) | 0 | OVF |
| | | $x_{l_{mi}}$ | 0.69 | 3 (3.2) | 0.92 | 2 (2.2) |
| OMPGS | P | original $x$ | 0.82 | 6 (6.3) | 0.94 | 2 (2.4) |
| | | $x_{l_{mi}}$ | 0.85 | 6 (6.1) | 1 | 2 (1.9) |
| | O | original $x$ | 0.54 | 6 (6.8) | 0.62 | 2 (2.8) |
| | | $x_{l_{mi}}$ | 0.71 | 6 (6.4) | 0.94 | 2 (2.1) |

Table 11: Pessimistic Robustness Assessment ($\Gamma=0$) after Model Robustness Enhancement by reducing $I(f_y;S)$. $\downarrow^{\mathbf{SR}}$ is the SR decreased amount comparing to the corresponding SR value with $\Gamma=0$ in Table 9.

| Model | Algo. | PEDec | | | EHR | | |
|---|---|---|---|---|---|---|---|
| | | SR | $\downarrow^{\mathbf{SR}}$ | Med (Avg) | SR | $\downarrow^{\mathbf{SR}}$ | Med (Avg) |
| AdvC | FSGS | 0.65 | 0.22 | 4 (4.2) | 0.94 | 0 | 2 (2.17) |
| | OMPGS | 0.58 | 0.24 | 6 (6.4) | 0.92 | 0.02 | 2 (2.82) |
| NuR | FSGS | 0.81 | 0.06 | 3 (2.9) | 0 | 0.94 | OVF |
| | OMPGS | 0.25 | 0.57 | 6 (6.1) | 0 | 0.94 | OVF |
| RS | FSGS | 0.93 | -0.05 | 3 (3.2) | 0.96 | -0.02 | 1 (1.97) |
| | OMPGS | 0.86 | -0.04 | 4 (4.1) | 0.91 | 0.03 | 2 (2.20) |
| **SRS** (Boj. et al., 2020) | | 1.0 | - | 1 | 1.0 | - | 1 |

the impact of feature informativeness on the robustness assessment scores at $\Gamma=0$ with the binary-class datasets *PEDec* and *EHR*. The table shows that the classifier becomes less robust to adversarial modifications over $x_{l_{mi}}$ with less predicative information.

### F.5.3 ROBUSTNESS ASSESSMENT AFTER ROBUSTNESS ENHANCEMENT

From the parameter of the decreased **SR** in Table 11, we study the robustness enhancement by different model enhancement techniques. On *PEDec*, the most robustness assessment scores increase slightly. Notably, the decreased **SR** of *AdvC* and *NuR* are positive and brings the most robustness improvement. On *EHR*, *NuR* can prevent attack over all the testing instances. As seen in Table 11, the robustness-enhancing techniques induce more visible impact on the pessimistic robustness assessment scores using **OMPGS**. This result meets the intuitive interpretations in the main paper.

