# OpenReview forum: "Towards Understanding the Robustness Against Evasion Attack on Categorical Data"
_ICLR.cc/2022/Conference — ICLR 2022 Poster_

### Official Review · Reviewer_W7KQ · 2021-11-02

**Correctness:** 4
**Technical Novelty And Significance:** 3
**Empirical Novelty And Significance:** 3
**Recommendation:** 6
**Confidence:** 4

**Main Review:**

Pros:
1. Different from the traditional analysis of continuous data, this paper focuses on the vulnerability of categorical data. This research has broad application scenarios for discrete data such as text analysis.
2. In the method part, the influence of I(x;y,S) and I(f;S) on model robustness is confirmed through rigorous theoretical analysis, and the effect of such influence is confirmed through quantitative data in the experiment part.
3. The theoretical derivation of the paper is rigorous. The authors set a theoretical association between the smoothness of the targeted classifier and the solvability of both pessimistic and optimistic robustness assessment with categorical inputs. In solving NP knapsack problem, provably accuracy is guaranteed.

After reading the paper I have some concerns as follow:
1. The author explains that the greedy search strategy is chosen because of its adaptability to different classifier architectures. However, I am still confused about the theoretical reasons for not using gradient method, after all, gradient method is more intuitive in solving problems. If the categorical input cannot be adapted through gradient calculation due to discrete data, what are the implementation details of gradient attack for Table 1? If gradient works, is the difference in performance due to the failure to find a better gradient attack, not necessarily the result of a greedy search strategy?
2. In Table 3, the author uses three defensive measures to verify the effect of reducing model and data set dependency on improving robustness. It restricts the reduction range of I(f;S) by controlling the robustness score equal to 0, but is this control method reasonable? For example, for nuclear norm regularization method, regularizing the model parameters during training can reduce I(f;S), but it will also affect other factors. Relying solely on constraining the final impact threshold result instead of constraining I, can we use this method of comparison to fairly reflect the difference in results?


**Summary Of The Paper:**

The paper mainly studies the adversarial vulnerability of a classifier with categorical inputs. It theoretically analyzes the key factors that determine the robustness of discrete input data and uses greedy strategy to solve the problem. In the experimental part, the importance of the key factors pointed out by the theory is verified, and the robustness scoring is used to conduct experiments under three defense measures.

**Summary Of The Review:**

In summary, the paper is novel, and gives some insights to the adversarial robustness.

---

### Official Review · Reviewer_FWFM · 2021-11-03

**Correctness:** 3
**Technical Novelty And Significance:** 3
**Empirical Novelty And Significance:** 3
**Recommendation:** 8
**Confidence:** 4

**Main Review:**

The idea of the paper is somewhat interesting: can we efficiently measure the model robustness to changes in categorical features? The paper explores both the pessimistic and the optimistic measurements. The empirical attack result is in between, which makes sense.

However, I have many questions regarding the paper organization and the impact of the results. The writing is really dense and lacks the link between the theoretical result and the experiment. Here are my main questions.

1) Theorem 1 measures the robustness over a specific training set S. Given the randomness of S, why is the statement deterministic? I was expecting something like "with high probability, the robustness given is random sample of S is at lease some value". What is the robustness in Theorem 1 with respect to? The underlying distribution or just over the training set?

2) The same question holds for other theorems too. Are the robustness measured on the training set or the data distribution?

3) The optimistic and pessimistic calculation of robustness seems to be heuristics too. Can we assert that they are close to the theoretical value as samples size increases? Or in other words, how do we know how well these estimators are? What's the advantage of using these approximations than just using a practical attack to probe?

4) Theorem 1 seems a bit off the big picture of the paper. There are definitely more important features than others. However, if a feature is really essential to determining the ground truth label, why would we expect robustness to perturbation over this feature? The semantics of the original input may have been changed already...

Please let me know your thoughts on these questions. Thank you.

In addition, please do not shrink the font size of an algorithm box unless you don't want the reviewers to read... The theory part is too dense unless they can be linked to finite sample cases.

**Summary Of The Paper:**

The paper proposes new methods to gauge model robustness to perturbations in categorical data. The experiments show the effectiveness of such new estimators and corroborates the intuition over model robustness v.s. mutual information over features.

=============================================
I acknowledge that I've read the author response. The response clarified my questions. Thanks for clarifying that the paper is not centered around defense mechanism but rather robustness validation. That also resolved some of my doubts over the practical use of it. I'm raising my score to 8.

**Summary Of The Review:**

Overall, I like the idea of gauging the robustness to changes in categorical features. However at this moment, I feel the setting is slightly artificial and the tightness of the gauge does not come across. (For example, can we estimate the value of $m_{\Omega}$ and $M_{\Omega}$?)

I feel the paper can be greatly improved by dropping the mutual information part. It's not contributing to the core ideas.

Therefore, I'm giving a score of 3 for now. I hope the authors can address my questions.

---

### Official Review · Reviewer_SsEb · 2021-11-03

**Correctness:** 3
**Technical Novelty And Significance:** 3
**Empirical Novelty And Significance:** 3
**Recommendation:** 6
**Confidence:** 3

**Main Review:**

> In general I think this paper is clearly written. Discrete inputs' adversarial robustness problem is not well-studied in this community but they are prevalent in language and biomedicine. The authors consider both the pessimistic and the optimistic scenario, which is also an interesting perspective.

> It seems to me that the theoretical results in this paper highly depend on the Lipschitz continuity of the model. This is not surprising and many prior works have shown the connection between adversarial robustness and the model's local Lipschitz. However, it is not clear to me how the Lipschitz number of models in the experiment looks like. I don't think currently we have a good method to accurately evaluate a DNN model's local Lipschitz. I think it would be great if the authors can provide some discussions on this.

> I think it would be better to provide a time complexity analysis and running time comparison of the proposed method. Also, for the greedy approach, in practice, would the loop break early in most cases?

**Summary Of The Paper:**

This paper studies the problem of assessing the adversarial robustness of a classifier with categorical inputs, instead of continuous inputs as in the literature. The authors claim that there exist provably optimality guarantees for Lipschitz-continuous classifiers, and proposed an impact factor based on an n information-theoretic analysis. Experimental studies are conducted to support the claims.

**Summary Of The Review:**

The problem studied is important and not well-studied in the literature. Some theoretical results are developed, though with strong assumptions. In general, I think this paper is interesting, so I recommend a weak acceptance and would be curious to see other reviewers' opinions as well as the authors' rebuttal.

---

### Decision · Program_Chairs · 2022-01-20

**Decision:**

Accept (Poster)

**Comment:**

In this manuscript, the authors study the relatively unexplored problem of how to characterize and assess the adversarial vulnerability of classification models with categorical input. Even certifying the robustness of such classification models is intrinsically an NP-hard combinatorial problem, the authors show that the robustness certification can be solved via an efficient greedy exploration of the discrete attack space for any measurable classifiers with a mild smoothness constraint.
Overall, the theoretical analyses in this paper are rigorous, and reviewers seem to be satisfied with the responses from the authors.
Based on the three positive reviewers, this manuscript is recommended to be accepted.